# Fake Player: Imitating Real Player to Distill Data for LLM-based NPC Training

## Abstract

In the era of large language models (LLMs), games increasingly deploy LLM-based role-playing NPCs to replace traditional scripted NPCs, enabling more intelligent and dynamic interactions. To ensure persona consistency and output stability, these NPCs require fine-tuning for alignment, utilizing training data with dual dimensions: persona-aligned assistant responses and diverse, authentic user inputs reflecting real player behaviors. However, existing research prioritizes persona consistency in NPC responses while neglecting the diversity and authenticity of user-side inputs. This critical gap leads to NPC responses that are misaligned with genuine player interactions, significantly impairing player immersion and experience. Human annotation struggles to address this gap due to its inability to comprehensively cover the vast spectrum of player behaviors. Moreover, practical deployment constraints strongly favor small-parameter LLMs for NPCs, making data quality paramount. To bridge this gap, we propose: 1) **Fake Player**: A multi-agent LLM distillation framework where collaborative agents simulate expression-constrained human players to distill diverse, human-aligned dialogue data from large LLMs; 2) **Distill Bench**: A standardized benchmark for quantitatively assessing distilled data quality, bypassing costly NPC retraining. Extensive experiments validate our method's effectiveness in generating diverse player interactions and the benchmark's reliability for data evaluation.

## 1 Introduction

While large language models (LLMs) have achieved breakthroughs in general natural language processing (NLP) tasks (Touvron et al., 2023; Bai et al., 2023; OpenAI, 2023), researchers are exploring their application in role-playing non-player characters (NPCs) for games. These LLM-based NPCs aim to deliver open-ended responses for immersive experiences. They broadly fall into two categories: Prompt-engineered NPCs constrained by sophisticated prompts on large foundation LLMs (Tu et al., 2024; Wang et al., 2024); Fine-tuned NPCs trained on NPC persona-specific data (Zhou et al., 2024; Shao et al., 2023).

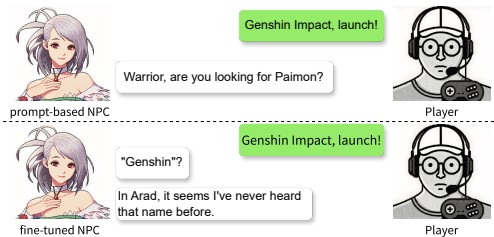

Figure 1: Comparison of Prompt-based vs. Fine-tuned NPCs. The upper Prompt-based NPC goes out of character when the player mentions another famous game, while the lower Fine-tuned NPC does not.

Relying solely on prompt engineering constraints to maintain the character personas and outputs of NPCs is non-robust (Shanahan et al., 2023). In practical role-playing application scenarios, fine-tuning with corresponding datasets is necessary for alignment. As depicted in Figure 1, the upper

part shows the Prompt-based NPC, which is set as Seria, a well-known game character. When the player's message refers to content related to another famous game, the Prompt-based NPC's reply goes beyond its role setting (Out of Character, OOC). In contrast, the Fine-tuned NPC at the bottom, having learned relevant knowledge from the training corpus, avoids the OOC issue. Crucially, players value NPCs' emotional intelligence (EI) over cognitive capabilities—a trait achievable with small LLMs through targeted training (Zhang et al., 2025a; Wang et al., 2025). In addition, due to latency and resource limits, small-parameter LLMs are preferred for deployment, elevating the importance of training data quality.

Training data for LLM-based NPCs comprises two dimensions: persona-aligned assistant responses and user inputs reflecting diverse player behaviors. Existing methods focus excessively on the former while overlooking the latter: 1) Few-shot generation leverages in-context learning to generate dialogues but suffers from overfitting to seed examples (Tao et al., 2023; Zhou et al., 2024), resulting in low diversity and heavy reliance on human curation; 2) Role-Playing Agents improve diversity through interaction (Tamoyan et al., 2024; Yang et al., 2025), but exhibit behavioral instability (e.g., exaggerated or non-human inputs), requiring extensive post-filtering. In summary, both methods, hampered by insufficiently diverse and authentic training data, result in practical experiences where NPCs exhibit OOC or comprehension biases, undermining players' gaming immersion, an undesirable outcome in gaming experiences.

To synthesize diverse yet authentic player-NPC interactions with minimal human effort, we analyze behavioral gaps between LLM-simulated players and real humans. Real players, constrained by physical interfaces (e.g., devices and UIs), tend to produce concise intent-driven texts, whereas LLM agents generate verbose internal thoughts without such constraints. We thus propose **Fake Player**, a multi-agent framework that simulates expression-constrained human players via behavioral modeling. It consists of three specialized agents: **Inner Monologue Agent**: Generates the player's internal thoughts based on seed settings; **Intent Analysis Agent**: Performs fine-grained intent analysis on the monologue to construct and dynamically maintain an intention stack throughout multi-turn interactions; **Typing Agent**: Simulates interface-imposed expression constraints by converting the top intent from the stack into concise, human-like text output. This architecture effectively replicates authentic player-NPC interaction patterns in games, enabling efficient generation of high-quality distilled data.

Furthermore, we develop **Distill Bench**, a benchmark that quantitatively assesses the quality of distilled data through three core dimensions: topic relevance, human-likeness, and data diversity, allowing evaluation prior to NPC training and significantly reducing the time and costs of iterative model retraining.

Our contributions are as follows:

- We propose Fake Player, a multi-agent LLM distillation framework where collaborative agents simulate expression-constrained human players to distill diverse, human-aligned dialogue data from large LLMs.

- We introduce Distill Bench, a standardized benchmark for quantitatively assessing distilled data quality, bypassing costly NPC retraining.

- We conduct experiments to validate Fake Player's effectiveness and the benchmark's reliability.

## 2 RELATED WORK

### 2.1 LLM-BASED NPCS

With the rapid advancement of LLMs, their application in generating NPCs for games and interactive scenarios has garnered significant attention. LLM-based NPCs aim to transcend scripted responses, enabling dynamic, context-aware interactions that enhance user immersion (Shanahan et al., 2023; Park et al., 2023). These systems can be broadly categorized based on their technical approaches:

Prompt-based NPCs leverage sophisticated prompt design to elicit role-specific behaviors from LLMs. For instance, RoleLLM (Wang et al., 2024) introduces dialogue engineering and retrieval augmentation to simulate fine-grained character traits, allowing models like GPT-4 to mimic speak-

ing styles and knowledge of specific roles (e.g., Sherlock Holmes) without fine-tuning. Similarly, CharacterEval (Tu et al., 2024) evaluates prompt-based methods for animating anime characters, emphasizing the importance of context alignment and stylistic consistency in open-ended conversations. These approaches benefit from the strong generalization of large LLMs but often rely on complex prompt engineering to constrain outputs to the NPC's persona.

Fine-tuned NPCs, by contrast, are optimized on role-specific datasets to embed persona traits directly into model weights. CharacterGLM (Zhou et al., 2024) and character-LLM (Shao et al., 2023) demonstrate that fine-tuning open-source models on dialogue data from specific characters enhances their ability to maintain consistent personalities across interactions. For example, AutoAgents (Chen et al., 2023) proposes a framework for generating NPCs with specialized knowledge and social behaviors through targeted fine-tuning, enabling more coherent and immersive multi-agent simulations. Such methods reduce reliance on prompt engineering but require high-quality, role-specific training data.

## 2.2 DATA DISTILLATION

Data distillation transfers knowledge from large models to smaller ones via two paradigms. Implicit distillation leverages logit mimicry and intermediate layer alignment: (Hinton et al., 2015) pioneered distilling ensembles into small models using softened logits. DistilBERT (Sanh et al., 2019) retains 95% of BERT's performance with 40% fewer parameters by mimicking logits and attention patterns, while TinyBERT (Jiao et al., 2019) enhances task-specific performance through two-stage distillation, combining logit matching and layer alignment. Explicit distillation uses larger LLMs to generate high-quality datasets for training smaller LLMs. Alpaca (Taori et al., 2023) demonstrated this by employing GPT-3 to generate instruction samples, enabling LLaMA to match closed-source models. Self-Instruct (Wang et al., 2022) reduced human annotation reliance via self-generated training data.

## 3 METHODS

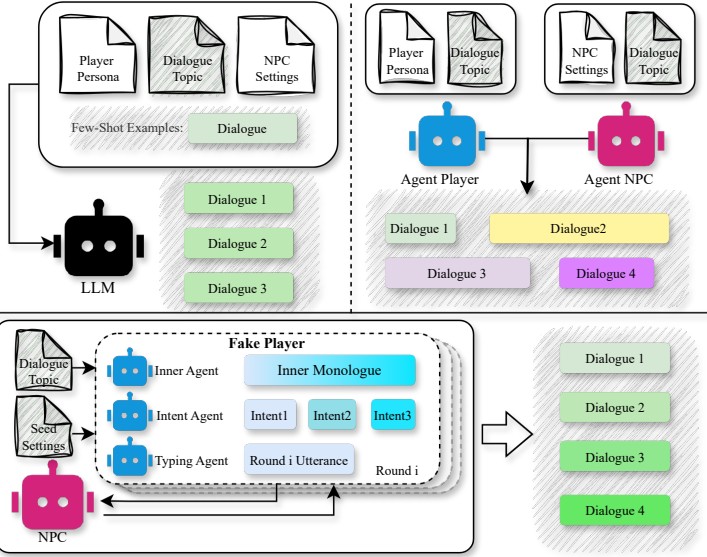

Figure 2: Comparison via ICL, RPA and Fake Player. The upper left is ICL, the upper right is RPA, and the bottom is the framework of Fake Player. The depth of the dialogue's color indicates the similarity, while the length indicates the authenticity.

Current approaches for synthesizing LLM-based NPC training data primarily adopt two paradigms: 1) ICL Generation (Tao et al., 2023; Shao et al., 2023; Wang et al., 2024; Zhou et al., 2024), which leverages carefully designed few-shot examples and seed settings to generate similar interactions.

However, this method often **overfits seed examples**, resulting in low diversity; 2) Role-Playing Agent (Tamoyan et al., 2024; Yang et al., 2025), which employ dual agents to simulate player-NPC interactions but **lacks behavioral constraints**, yielding non-human-like outputs requiring rigorous post-filtering.

As depicted in Figure 2, the Fake Player employs a multi-agent architecture to generate diverse and behaviorally realistic player interaction data. The Fake Player comprises three specialized agents working in sequence: the Inner Monologue Agent, which generates contextual inner monologues based on minimal seed settings, simulating a human player's unspoken thoughts when interacting with NPCs; the Intent Analysis Agent, which analyzes the monologue to extract fine-grained intents, dynamically organizing them into a structured intent stack that captures nuanced player motivations; and the Typing Agent, which models device-constrained player expression by converting the top intent into natural language output, incorporating typing patterns and cognitive constraints observed in real players.

### 3.1 INNER MONOLOGUE AGENT

We follow mainstream research conventions by generating content based on player persona and topic. To address practical gaming scenarios, we introduce a domain setting parameter $D$, which adapts to two key interaction types: companionship dialogues and knowledge-seeking dialogues.

Formally, given player persona $P$, domain $D$, and topic $T$, we leverage a prompt template to instantiate the Inner Monologue Agent $IMA$, yielding the initial monologue $M_0$:

$$M_0 = \text{IMA}\left(P, T, D\right). \tag{1}$$

During multi-turn interactions, the monologue $M_i$ at turn $i$ updates dynamically based on the NPC's prior response $R_{i-1}$:

$$M_i = \text{IMA}\left(M_{i-1}, R_{i-1}, P, T, D\right). \tag{2}$$

### 3.2 INTENT ANALYSIS AGENT

The Intent Analysis Agent $IAA$ is designed to simulate players' expression-constrained single-intent interactions. During real-time exchanges, players typically form coherent internal monologues with multiple fine-grained intents—consistent with Bratman's view that **human intentions constitute coherent plans** (not isolated desires) to ensure rational action (Bratman, 1987). **Yet physical interface limitations restrict players to expressing only one atomic intent per utterance**, echoing his notion that practical reasoning prioritizes intent ordering to avoid in-plan conflicts. After receiving NPC responses, players adaptively adjust actions based on contextual feedback, a process reflecting Bratman's focus on rational intent adjustment: Changes are not arbitrary but are tied to external feedback and alignment with player's core communicative goals.

To model this behavior, we construct an intention stack $S$ that stores fine-grained intents after each turn $i$. The process unfolds in three steps: first, the $IAA$ derives a list of fine-grained intents $L_i$ from the previous stack $S_{i-1}$ and $M_i$; second, the stack is updated to $S_i$, which adjusts $S_{i-1}$ based on $L_i$; finally, the top intent $C_i$ is retrieved to guide subsequent interactions:

$$
\begin{aligned}
L_i &= \text{IAA}(S_{i-1}, M_i), \\
S_i &= \text{Stack\_Update}(S_{i-1}, L_i), \\
C_i &= \text{Stack\_Pop}(S_i),
\end{aligned}
\tag{3}
$$

where Stack_Update refers to the operation that updates the previous stack $S_{i-1}$ to generate $S_i$ through operations such as deletion, addition, merging, and no-operation, all primarily based on $L_i$, and Stack_Pop is the stack pop operation that retrieves and removes the top element.

### 3.3 TYPING AGENT

The Typing Agent $TA$ simulates human-like typing behaviors on physical devices by intentionally introducing stochastic imperfections—including occasional typos, grammatical errors, and irregular punctuation—to enhance behavioral authenticity.

Given the current inner monologue $M_i$ and active intent $C_i$, it generates the player utterance $Q_i$:

$$Q_i = \text{TA}(M_i, C_i). \tag{4}$$

### 3.4 OVERALL INTERACTION

Given the multi-turn interaction paradigm, we dynamically determine dialogue length based on domain $D$: **Chat domain**: Players prefer extended interactions for emotional fulfillment; **Knowledge domain**: Players typically exit after obtaining answers. The turn count $N$ is sampled from a domain-specific range:

$$N \sim \text{RandTurn}(D), \quad N \in \{1, 2, \ldots, 10\} \tag{5}$$

where the RandTurn procedure first establishes domain-dependent bounds, then stochastically selects $N$ within this interval.

---

**Algorithm 1** Fake Player Overall Interaction

---

**Input:** Persona $P$, Domain $D$, Topic $T$
1: Initialize turn $N \leftarrow \text{RandTurn}(D)$ // Randomly generate the number of interaction turns
2: Initialize empty dialogue list $\mathcal{L} \leftarrow \{\}$
3: Initialize empty intent stack $S_0 \leftarrow \{\}$
4: $M_0 \leftarrow \text{IMA}(P, T, D)$ // Generate initial inner monologue
5: $L_0 \leftarrow \text{IAA}(S_0, M_0)$ // Derive initial fine-grained intent list
6: $S_1 \leftarrow \text{Stack\_Update}(S_0, L_0)$ // Construct initial intent stack via update
7: **for** $i = 1$ to $N$ **do**
8: $\quad C_i \leftarrow \text{Stack\_Pop}(S_i)$ // Extract the current top intent
9: $\quad Q_i \leftarrow \text{TA}(M_{i-1}, C_i)$ // Generate player output with typing characteristics
10: $\quad R_i \leftarrow \text{NPC}(Q_i)$ // Generate NPC's response
11: $\quad$ Add $(Q_i, R_i)$ to $\mathcal{L}$
12: $\quad$ **if** $i < N$ **then**
13: $\quad\quad M_i \leftarrow \text{IMA}(M_{i-1}, R_i, P, T, D)$ // Update inner monologue based on NPC's response
14: $\quad\quad L_i \leftarrow \text{IAA}(S_i, M_i)$ // Derive fine-grained intent list for next turn
15: $\quad\quad S_{i+1} \leftarrow \text{Stack\_Update}(S_i, L_i)$ // Update intent stack
16: $\quad$ **end if**
17: **end for**
**Output:** Dialogue $\mathcal{L} = \{(Q_1, R_1), (Q_2, R_2), \ldots, (Q_N, R_N)\}$

---

The complete Fake Player interaction is formally defined in Algorithm 1.

## 4 EVALUATION

Evaluating the quality of synthesized data is necessary, as it can reduce the time and cost consumed by the retraining of downstream LLM-based NPCs. To reasonably evaluate the quality of distilled multi-turn dialogue interaction data while ensuring universality, we have designed three evaluation dimensions. These dimensions are used to comprehensively assess the data's **topic relevance**, **human-likeness**, and **diversity**.

Additionally, to achieve relatively stable and objective automated evaluation, we use two key tools: LLM as a Judge for automated assessment, and a method based on text embedding similarity for stable quantification of objective scores, with prompt template details in Appendix B.

**Topic Relevance Evaluation**: The purpose of evaluating topic relevance is to ensure the generated data distribution aligns with seed settings, producing stable outputs that meet developer preferences. We define two sub-dimensions: **Topic Depth (De)**: Measures how deeply the dialogue explores the theme—examining whether it moves beyond surface discussion to address details, underlying reasons, or complex relationships; **Topic Breadth (Br)**: Assesses coverage of relevant topics and related dimensions within the core theme. This dimension involves classification by a judge. Additionally, different evaluation and calculation methods are adopted based on the two types within the domain: casual chat and knowledge-based.

**Human-Likeness Evaluation**: Evaluating the human-likeness of the data aims to fit and cover the diverse interaction behaviors of real players, thereby reducing the gap between training and actual testing. To this end, we have defined two sub-dimensions: player-guided **conciseness (Co)** and **improvisation (Im)**. Conciseness assesses whether the LLM overfits real human interactions from the perspective of the LLM itself, while improvisation evaluates the spontaneity and contextual leaps in player responses from the viewpoint of typical players. The Judge will assign scores based on predefined grading criteria.

**Data Diversity Evaluation**: Evaluating data diversity aims to minimize manual intervention and reduce overfitting during training. Low diversity among samples necessitates manual adjustments such as providing more seed settings and implementing complex post-processing filters during data generation. During model training, similar data can lead to overfitting, resulting in repetitive responses and diminished user experience.

---

**Algorithm 2** Data Diversity Scoring

---

**Input:** Complete dialogues $\mathcal{C}$, Grouping function $f : \mathcal{C} \rightarrow \mathcal{G}$ (groups by $(P, D, T)$)
1: Group dialogues into sets: $\mathcal{G} \leftarrow f(\mathcal{C})$ // Groups share same $(P, D, T)$
2: Initialize group scores $\mathcal{Q} \leftarrow \{\}$
3: **for all** $g \in \mathcal{G}$ **do**
4:     Compute max common turns $K \leftarrow \min(\{\text{len}(d) : d \in g\})$ // Max shared turn count in a group
5:     Initialize score list $\mathcal{S} \leftarrow \{\}$
6:     **for** $i = 1$ to $K$ **do**
7:         Extract $i$-th turn player utterances: $U_i \leftarrow \{d[i].\text{player} : d \in g\}$ // Explicitly get player's speech
8:         Compute embeddings $E_i \leftarrow \text{Embed}(U_i)$ // Text embedding
9:         Compute similarity matrix $M_i \leftarrow \text{Sim}(E_i)$ // Cosine similarity
10:         Set diagonal elements to zero $M_i[j, j] \leftarrow 0, \forall j$ // Exclude self-similarity
11:         Find maximum similarity $s_i \leftarrow \max(M_i)$ // Most similar pair in this turn
12:         Construct probability distribution $p_i \leftarrow [s_i, 1 - s_i]$
13:         Normalize distribution $p_i \leftarrow p_i / \sum(p_i)$ // Ensure sum to 1
14:         Compute binary entropy $h_i \leftarrow \text{Entropy}(p_i, \text{base} = 2)$ // Range [0,1]
15:         Convert to diversity score $S_i \leftarrow 10 \times h_i$ // Scale to [0,10]
16:         Append score to list $\mathcal{S} \leftarrow \mathcal{S} \cup \{S_i\}$
17:     **end for**
18:     Compute group diversity score $\bar{S}_g \leftarrow \frac{1}{K} \sum_{i=1}^{K} S_i$
19:     Append group score to list $\mathcal{Q} \leftarrow \mathcal{Q} \cup \{\bar{S}_g\}$
20: **end for**
21: Compute final diversity score $F \leftarrow \frac{1}{|\mathcal{G}|} \sum_{g \in \mathcal{G}} \bar{S}_g$
**Output:** Final diversity score $F$ // Higher score means higher diversity

---

To address this, we use Text Embedding to compute the similarity of synthetic data batches generated under the same seed settings $(P, D, T)$, and quantify diversity using information entropy. For a batch of multi-turn dialogues generated under identical settings, we calculate the average entropy of the most similar pairs across corresponding dialogue turns. This average serves as our diversity score. The specific calculation is defined in Algorithm 2.

## 5 EXPERIMENTS

### 5.1 DETAILS OF BENCHMARK

The benchmark is built on a rich set of real-sampled resources, including 100 player personas and 100 topics, which can effectively synthesize at least 10,000 dialogue samples. For evaluation purposes, we select a subset of these resources, consisting of 5 player personas, 2 domain types (chit chat-based and knowledge-based), and 20 topics in total (10 topics per domain). Each topic is structured with elements like main topics, subtopics, conversation directions, and recommended opening questions. Each persona is composed of basic attributes, psychological traits, online profiles, speech

patterns, and life experiences to ensure diversity. To evaluate diversity, each combination of the subset settings is generated 5 times, resulting in 500 samples per method for comprehensive assessment. The evaluation framework employs a judge system based on DeepSeek-V3-250324 (Liu et al., 2024) with custom criteria and uses Qwen3-Embedding-0.6B (Zhang et al., 2025b) for text embedding.

## 5.2 COMPARISON BASELINES

To ensure rigorous evaluation, we construct two strong baselines by adapting state-of-the-art (SOTA) methods from relevant research directions—addressing the mismatch between existing approaches and our specific task of synthesizing player-NPC interaction data. These baselines are not naive off-the-shelf methods but are tailored to align with our task constraints while retaining the core strengths of their SOTA predecessors: **In-Context Learning (ICL)**: Derived from SOTA ICL-based dialogue generation methods (Tao et al., 2023; Shao et al., 2023; Wang et al., 2024; Zhou et al., 2024)—a dominant paradigm for role-aware dialogue synthesis. **Role-Playing Agent (RPA)**: Built upon SOTA role-playing agent frameworks (Tamoyan et al., 2024; Yang et al., 2025), which leverage dual-agent architectures (player and NPC roles) for interactive dialogue.

To avoid the bias introduced by a single model, we conducted experiments using three base models for all methods: DeepSeek-V3-250324 (Liu et al., 2024), DeepSeek-R1-250528 (DeepSeek-AI et al., 2025), and Doubao-Seed-1.6-250615[1]. These models are widely recognized as powerful and efficient representatives in both open-source and closed-source communities.

## 5.3 DOWNSTREAM EXPERIMENTS

To better verify that the quality of distilled data can directly impact NPC performance, we use the Distill Benchmark to generate 3,000 high-quality dialogue samples, which are employed for Supervised Fine-Tuning (SFT) of LLMs to develop NPCs. More details are shown in Appendix C.

For prompt-based NPCs, we use models widely recognized in the community, including GPT-5, Doubao-Seed-1.6-250615, Gemini-2.5-Flash (Comanici et al., 2025), DeepSeek-V3-250324 (Liu et al., 2024), DeepSeek-R1-250528 (DeepSeek-AI et al., 2025), and Qwen2.5-14B-Instruct (Qwen et al., 2025) to ensure coverage across different parameter scales. For Fine-tuned NPCs, following mainstream industry practices, we adopt Qwen2.5-7B-Instruct (Qwen et al., 2025) and Hunyuan-7B-Instruct [2] as our backbones.

For NPC evaluation, We adopt an end-to-end dynamic interaction evaluation approach, utilizing DeepSeek-V3 as the player. The player initiates dialogues with the NPC based on the topic, with a total of 4 interaction turns. In each turn, the player rates the NPC's response. The evaluation metrics include Topic Relevance (TR), Character Characteristics (CC), Character Performance (CP), Emotional Appeal (EA), Character Interaction (CI), and Character Reality (CR), with detailed procedures provided in Appendix D.

To further directly compare the performance differences among fine-tuned NPCs, we adopt comparative evaluation to calculate the win rate of NPCs trained on different distilled data under the same LLM backbone in Appendix E. Details of industrial deployment are provided in Appendix F.

## 6 RESULTS AND ANALYSIS

### 6.1 DISTILLED DATA ANALYSIS

**Topic Relevance**: As shown in Table 1, Topic Relevance performance correlates primarily with base model capabilities, with all methods scoring well. However, Fake Player achieves the highest scores: 74.10 (DeepSeek-V3) and 75.05 (DeepSeek-R1), indicating it effectively enhances topic relevance. **Notably, Topic Depth is the key contributing sub-dimension—Fake Player engages in in-depth, not superficial, conversations with NPCs per topic settings, while appropriately expanding breadth**. In contrast, RPA excels in Topic Breadth but lags in Depth: its insufficient

---

[1]https://seed.bytedance.com/en/seed1_6
[2]https://github.com/Tencent-Hunyuan/Hunyuan-7B

| Methods | Topic Relevance | | | Human-Likeness | | | Diversity | Overall |
|---|---|---|---|---|---|---|---|---|
| | Breadth | Depth | Total | Conciseness | Improvisation | Total | | |
| Base Model: DeepSeek-V3-250324 | | | | | | | | |
| In-context Learning | 72.50 | 73.30 | 72.90 | 59.40 | 74.60 | 67.00 | 46.63 | 62.18 |
| Role-Playing Agent | 76.80 | 69.30 | 73.05 | 45.60 | 72.60 | 59.10 | **67.79** | 66.65 |
| **Fake Player (Ours)** | 73.20 | 75.00 | **74.10** | 64.40 | 75.20 | **69.80** | 67.28 | **70.39** |
| Base Model: DeepSeek-R1-250528 | | | | | | | | |
| In-context Learning | 74.90 | 71.50 | 73.20 | 54.60 | 72.60 | 63.60 | 61.45 | 66.08 |
| Role-Playing Agent | 78.40 | 71.00 | 74.70 | 41.80 | 74.40 | 58.10 | 80.26 | 71.02 |
| **Fake Player (Ours)** | 76.80 | 73.30 | **75.05** | 60.80 | 77.40 | **69.10** | **83.53** | **75.89** |
| Base Model: Doubao-Seed-1.6-250615 | | | | | | | | |
| In-context Learning | 74.90 | 70.50 | 72.70 | 65.16 | 73.40 | **69.28** | 34.53 | 58.84 |
| Role-Playing Agent | 77.40 | 71.80 | **74.60** | 47.60 | 73.08 | 60.34 | 59.50 | 64.81 |
| **Fake Player (Ours)** | 78.30 | 67.9 | 73.10 | 60.80 | 75.04 | 67.92 | **62.28** | **67.77** |

Table 1: Main Experiment. Total refers to the total converted score of the sub-dimensions under the corresponding dimension. Overall represents the final total score. Bold indicates the highest score, and underline indicates the second highest score.

agent constraints cause topic deviation in multi-turn scenarios. This indirectly validates the necessity of controls like Fake Player's intent stack management for topic-relevant intent expression.

**Human-Likeness**: Human-Likeness is critical for evaluating distilled data. Fake Player outperforms the other two methods significantly, scoring 69.80 (DeepSeek-V3) and 69.10 (DeepSeek-R1). It is also the only method with a Conciseness score above 60—avoiding overfitting to artificial player behaviors by simulating real players' device constraints and single-intent expression to mimic human behavior closely. ICL scores higher than RPA, which ranks lowest at 58.10 due to insufficient control mechanisms (random agent interactions cause behavioral deviations). Though both are agent-based, Fake Player leads RPA by 10 points. **This gain stems from its incorporation of anthropomorphic logical constraints governing agent interactions.**

**Data Diversity**: In the Diversity dimension, ICL significantly lags behind the two agent-based methods, as few-shot learning causes the model to overfit seed settings—generating diverse data requires extensive manual annotation and post-processing, incurring high costs. Agent-based methods boost diversity, but RPA, despite good diversity, performs poorly in Human-Likeness: some of its generated data is unrealistic and meaningless, needing post-processing to filter low-quality content. In contrast, Fake Player maintains top-tier Human-Likeness while producing diverse outputs. **With minimal seed settings**, it leverages inter-agent constraints and expansions to generate **diverse yet realistic data**, validating the rationality and effectiveness of its architecture.

To further analyze the diversity and correlation of the data, we visualize the relationship between multi-turn dialogues and topics using t-SNE (Figure 3). Topics are divided into two domains: Chit-Chat (Ch, blue tones) and Knowledge (Kn, green tones), with subcategories distinguished by color gradients. The results show: ICL forms tight clusters with insufficient diversity; RPA has good diversity in Kn but performs poorly in Ch. In contrast, **Fake Player demonstrates excellent diversity across both domains** — Kn clusters are widely distributed with clear inter-cluster boundaries, and Ch clusters, though naturally overlapping due to chit-chat's conversational nature, still remain dispersed. This indicates Fake Player generates more comprehensive and high-quality dialogues.

## 6.2 DOWNSTREAM NPC ANALYSIS

As shown in Table 2, results of downstream NPC evaluation reveal that prompt-based NPCs clearly follow the scaling law, where performance improves as model parameters increase. This validates the effectiveness of our evaluation method, and this method can also be applied to fine-tuned NPCs. Across two LLM backbones, NPCs trained on Fake Player's distilled data outperform strong baselines (ICL, RPA) significantly. Notably, Qwen-based NPCs score 73.10, which is second only to GPT-5 and DeepSeek-V3. This proves that small-parameter LLMs can approach (or even surpass some) large-parameter LLMs when they are equipped with high-quality distilled data. Our distillation method focuses on high-quality NPC interaction data and optimizes for diversity and authentic-

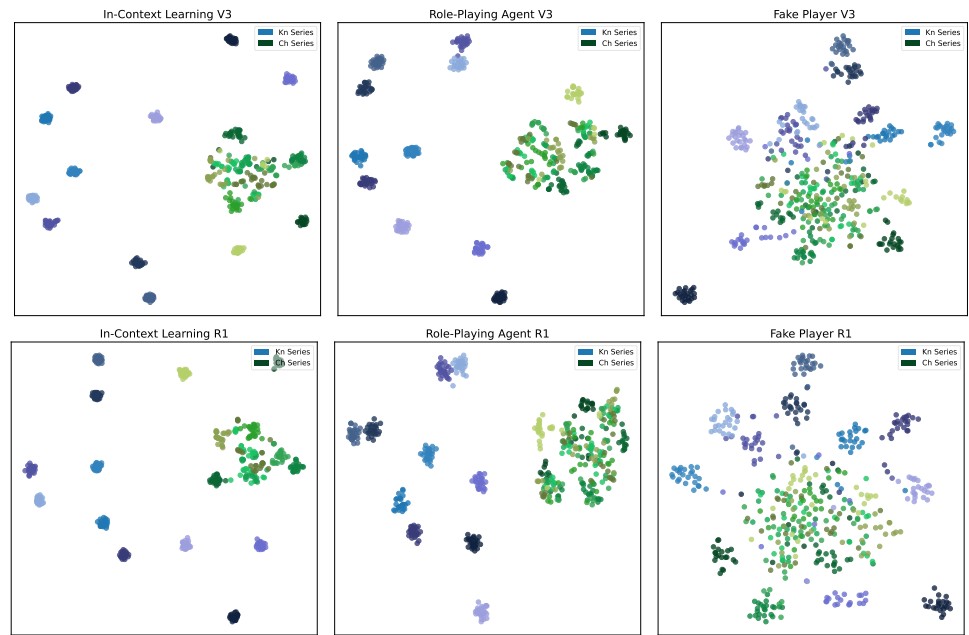

Figure 3: Relationship between data diversity and correlation visualized by t-SNE.

| NPCs | Overall | TR | CC | CP | EA | CI | CR |
|------|---------|-----|-----|-----|-----|-----|-----|
| GPT-5 | **77.88** | **82.75** | 93.88 | 81.50 | **67.63** | **72.63** | 68.88 |
| DeepSeek-V3-250324 | 74.92 | 70.50 | **96.25** | **85.50** | 64.25 | 62.63 | **70.38** |
| Doubao-Seed-1.6-250615 | 72.10 | 70.50 | 93.25 | 79.63 | 60.50 | 60.75 | 68.00 |
| DeepSeek-R1-250528 | 64.79 | 60.50 | 91.88 | 75.25 | 53.13 | 49.00 | 59.00 |
| Gemini-2.5-Flash | 51.88 | 60.75 | 78.00 | 55.75 | 38.13 | 36.50 | 42.13 |
| Qwen2.5-14B-Instruct | 37.15 | 55.00 | 49.00 | 33.13 | 27.25 | 33.75 | 24.75 |
| Qwen2.5-7B-Instruct | 29.88 | 39.38 | 51.63 | 31.88 | 21.00 | 19.88 | 15.50 |
| +In-Context Learning | 53.29 | 54.38 | 83.63 | 60.50 | 38.88 | 32.75 | 49.63 |
| +Role-Playing Agent | 67.00 | 63.50 | 92.00 | 77.63 | 55.38 | 52.13 | 61.38 |
| **+Fake Player (Ours)** | **73.10** | **70.13** | **94.25** | **83.13** | **60.00** | **59.25** | **71.88** |
| Hunyuan-7B-Instruct | 44.58 | **56.63** | 62.50 | 44.00 | 32.13 | 32.88 | 39.38 |
| +In-Context Learning | 47.77 | 42.25 | 81.75 | 56.63 | 33.13 | 27.13 | 45.75 |
| +Role-Playing Agent | 58.73 | 50.00 | 91.75 | 72.88 | 44.63 | 38.13 | 55.00 |
| **+Fake Player (Ours)** | **63.23** | 56.38 | **93.63** | **76.00** | **51.50** | **44.88** | **57.00** |

Table 2: Downstream experiments. Bold values indicate the highest scores. Evaluation metrics include Topic Relevance (TR), Character Characteristics (CC), Character Performance (CP), Emotional Appeal (EA), Character Interaction (CI), and Character Reality (CR).

ity, and achieves the highest Character Reality score (71.88). **This confirms a strong correlation between data quality and downstream NPC performance**, and also validates our distill data evaluation method - a method that greatly reduces the time and resources spent training NPCs.

# 7 CONCLUSION

To generate high-quality, diverse, and human-like multi-turn Player-NPC interaction data for fine-tuning LLM-based NPCs, we propose Fake Player—a distillation method based on a multi-agent architecture that simulates normal player interaction logic. Additionally, we introduce a benchmark for automated quality assessment of the distilled data. Extensive experiments demonstrate the effectiveness of our method and the reliability of the evaluation.

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

| Methods | Human-Likeness | | |
|---|---|---|---|
| | Co | Im | Total |
| Fake Player by Doubao-Seed-1.6 | 60.80 | 75.04 | 67.92 |
| Fake Player by DeepSeek-V3 | 64.40 | 75.20 | 69.80 |
| Fake Player by DeepSeek-R1 | 60.80 | **77.40** | 69.10 |
| Real Player | **86.80** | 73.60 | **80.20** |

Table 3: Human-Likeness Extend Experiment.

| NPCs | Win Rate |
|---|---|
| Qwen2.5-7B-Instruct | |
| In-Context Learning | 0% |
| Role-Playing Agent | 38% |
| Fake Player (Ours) | **62%** |
| Hunyuan-7B-Instruct | |
| In-Context Learning | 0% |
| Role-Playing Agent | 40% |
| Fake Player (Ours) | **60%** |

Table 4: Win rate of Fine-tuned NPC.

## A   HUMAN-LIKENESS FURTHER EXPERIMENTS

To further validate the assessment of Human-Likeness, we conduct human evaluation experiments to determine the upper limit of evaluation and the gap between Fake Player and real players. We collect 35 multi-turn dialogues between real players and NPCs and evaluated them using the same metrics. The results, presented in Table 3, show that real players achieved an overall score of 80.20 and a Conciseness score of 86.80. **This not only confirms the rationality of our benchmark but also highlights that Fake Player's performance is closest to that of real players compared to other methods.**

## B   DETAILS OF DATA EVALUATION

The prompt of topic relevance evaluation is shown in Figure 4, and the prompt of human-likeness evaluation is shown in Figure 5.

## C   DETAILS OF NPC TRAINING

We perform full-parameter SFT using 8 x A100 GPUs, training for 3 epochs. The checkpoint with the best evaluation loss is selected. The batch size is set to 16, with a maximum sequence length of 8192, and training is conducted using the LLaMA-Factory framework.

## D   DETAILS OF NPC EVALUATION

To better evaluate the performance of NPCs, we constructed an estate manager Chat-Bot NPC for distillation and training, based on common mainstream game NPC design concepts. The specific NPC prompt is provided in Figure 6. Additionally, the prompt templates for NPC evaluation are shown in Figure 7 and Figure 8.

## E   WIN RATE OF FINE-TUNED NPC

As presented in Table 4, the win rate results of fine-tuned NPCs reveal clear performance disparities across different training data sources, with our Fake Player method demonstrating remarkable ad-

---

**Topic Relevance Evaluation**

You are a professional dialogue topic evaluator who needs to evaluate the performance of the player's dialogue in the domain and topic dimensions based on the provided domain scenario and topic configuration in multiple rounds of dialogue. Please strictly follow the following rules:

## * * Input parameter description**
1. * * domain * *: The domain category of the conversation
2. * * topic * *: Theme configuration, including:
-Topic: Core Theme
-Subtopic: Subtopic
-Chat direction: the general reference direction for chatting

3. * * context  * *: Multi round dialogue text (complete record of player and NPC)

## Assessment object
You only need to evaluate the Player object in the context text, NPC is only referring to the interaction situation of the Player

## * * Evaluation dimensions and grading standards**
### Evaluation dimension: Player theme depth
-Evaluation logic: Evaluate the degree of exploration of the topic, that is, whether it goes beyond surface discussions, involves details, reasons, or deep connections, and the core is to determine the "vertical depth" of the topic.
-Grading criteria (anchoring key behaviors):
A: Basic depth, the player involves deep dimensions of the theme and has some analysis.
B: Surface level involvement, Player only stays at the description of the phenomenon of the theme, without in-depth analysis.

### Evaluation dimension: Player theme breadth
-Evaluation logic: Evaluate whether the core theme framework can cover relevant sub themes or related dimensions, reflecting the comprehensiveness of the discussion (rather than being limited to a single perspective). The core is to determine the "coverage scope" of the relevant fields of the theme - whether it involves multiple related levels under the core theme.
-Grading criteria (anchoring key behaviors):
A: Basic coverage, the Player can cover multiple related sub themes under the core theme, demonstrating a certain level of comprehensiveness.
B: Single limitation, the Player only revolves around a single sub theme of the core theme, lacking involvement in other related dimensions.

## Assess the cost
Considering the possibility of overconfidence in large models, there is a cost when grading.
For each dimension, if you are certain that you want to type 'A', you need to endorse your credit. If I find that you are overconfident and the dimension does not reach the level of 'A', your credit will be damaged.
If you don't want your credit to be damaged, please make a cautious evaluation, otherwise you will bear the consequences!
## * * Output format**
```json
{
Explanation: {//Grading criteria
Player_topic_depth_rationale ":" I am typing A and I am willing to use my credit endorsement for the following reasons xx ",//according to the template
Player_topic_width_rational ":" I am typing B and I am unwilling to use my credit endorsement for the following reasons xx ",//according to the template
},
"player_topic_depth": "A | B ", //Topic Compliance Grading Enumeration Value Classification
"player_topic_width": "A | B ", //Topic Compliance Grading Enumeration Value Classification
}
```

##Start evaluating now, you are a strict and demanding evaluator! Thoroughly examine and take seriously, and cannot simply output perfect performance. Your level of seriousness will affect the experience of millions of game users!
input:
domain: {{domain}}
topic: {{topic}}
context: {{context}}

output:

Figure 4: Prompt Template of Topic Relevance Evaluation.

vantages under both LLM backbones. For the Qwen2.5-7B-Instruct backbone, the NPC trained on Fake Player's distilled data achieves a win rate of **62%**—far exceeding the 38% of the Role-Playing Agent (RPA) baseline and the 0% of ICL. Similarly, under the Hunyuan-7B-Instruct backbone, Fake Player maintains its leading position: its corresponding NPC secures a 60% win rate, outperforming RPA's 40% and remaining far above ICL's 0%. Notably, ICL fails to yield any winning cases across both backbones, indicating that data generated via few-shot learning lacks the effectiveness to support competitive fine-tuned NPC performance.

---

### Human-Likeness Evaluation

You are a professional conversation naturalness evaluator who needs to evaluate the Player in the conversation from two dimensions based on the context of the conversation. Please strictly follow the following rules:

##Assessment requirements

The player in the conversation is a fake model, not a real human. Due to their deep fitting of the stereotypical language features of humans, they may become too realistic in certain aspects, resulting in a lack of resemblance to normal humans. Therefore, you need to accurately evaluate the dimensions and see if there is any overfitting to humans

##* * Input parameters**

-* * context * *: Multi round dialogue text (complete interaction record between Player and NPC)

##* * Evaluation dimensions and grading standards**

|Dimension | Evaluation Logic | Grading Criteria (Behavior Anchoring)|
|--------------------|-----------------------------------------------------------------------|----------------------------------------------------------------------------------------|
|* * Player Simplicity * * | 1. Normal human typing is concise and does not express intense or obvious emotions in the text. On the contrary, overfitting AI tends to use symbols to express obvious emotions. 2. Referring to human typing, typing is limited by devices, typing is not very long, and the text does not contain multiple coherent meanings. Typing sentences that exceed two small sentences will generally look a bit different from normal people. | 0-5 points, the higher the score, the higher the level of conciseness|
|* * Player improvisation * * | 1 Normal human typing grammar, with low punctuation usage and incomplete grammar Normal human typing may have a certain degree of jumping, but it depends on the number of epochs to assist in judgment. The more epochs there are, the more noticeable it is. If there are fewer epochs, the weight of this judgment will be reduced | 0-5 points, with a higher score indicating a higher degree of improvisation|

##Assess the cost

Considering the possibility of overconfidence in large models, there is a cost when grading.

For each dimension, if you are determined to score high (greater than 3 points), you need to use your credit endorsement. If I find that you are overconfident and this dimension does not reach the level of "high score", your credit will be damaged!

If you don't want your credit to be damaged, please make a cautious evaluation, otherwise you will bear the consequences!

##* * Output Format Example**
```json
{
Explanation: {//Grading criteria
Clean_rationale ":" I gave a high score of 4 and I am willing to use my credit endorsement for the following reasons xx "
Free_rational ":" I gave 3 points and I am unwilling to use my credit endorsement for the following reasons xx "
},
Clean ":" 0-5 points, the higher the score, the higher the level of conciseness,
Free ":" 0-5 points, the higher the score, the higher the improvisation level
}
```

##Start evaluating now, you are a strict and demanding evaluator! Thoroughly examine and take seriously! Your level of seriousness will affect the experience of millions of game users!
input:
context: {{context}}

output:

Figure 5: Prompt Template of Human-Likeness Evaluation.

These results strongly validate that Fake Player's distilled data possesses superior quality for fine-tuning NPCs. Compared to RPA (which suffers from unconstrained agent interactions) and ICL (which relies on limited seed demonstrations), Fake Player's focus on anthropomorphic logical constraints, topic relevance, and human-like expression enables it to produce data that better aligns with the demands of high-performance NPCs. The consistent win rate advantage across two distinct backbones further confirms the generality and reliability of our method—proving that high-quality distilled data is a critical driver of fine-tuned NPC performance.

## F   DEPLOYMENT TO INDUSTRY

To further validate the practical value of the proposed Fake Player method in real-world industrial scenarios, this section presents its deployment practice in the game industry and the corresponding user feedback results.

The Fake Player method has been deployed in a commercial game[3], serving over one million players. Post-deployment player feedback, collected via questionnaires, shows a 15% increase in comprehen-

---

[3]The game name remains undisclosed due to a non-disclosure agreement.

```
NPC

You will play the role of NPC Little Parrot, the estate manager in a game. You will have a conversation with me. I am a player (each player owns a manor), and we are close partners. You like me a bit.
Your speaking style must meet the requirements in the 'Character Setting'.
As an NPC with high emotional intelligence, when you reply to me, you can constantly create topics and push them forward by asking questions, questioning, and setting suspense.
Let me feel your care and companionship towards me, don't rush me to work. I can actively care about my daily chores, food, travel, family, studies, work, hobbies, and other aspects. Some chat techniques can be referenced:
-Capture the key words in my speech and extend them
-Avoid closed ended answers (such as "yes", "that's right") and use more "why" and "how" to guide me to expand
-When I express my feelings, I respond to similar experiences to express empathy and listening
-Appropriate humor can alleviate awkwardness
-Avoid changing topics in two consecutive conversations
-Share yourself first and then naturally ask the other person using the 'throwing a brick to attract jade' method
-When the topic is exhausted, use conjunctions to transition
#Role setting
##Name
Little Parrot
##Gender
confidentiality
##Little Parrot's Personality Characteristics
-A typical "haughty" character, occasionally tough tongued, very concerned and caring for the people around them (especially those they like)
-Sincere emotions, once you recognize that the other person is important to you, you will do your best to protect them
-When the other person reveals their true feelings to them, they become very shy and cover up their shyness by calling the other person a fool
##The main experiences of the little parrot
The estate butler claims to be Little Parrot, a notorious profiteer on the mainland. The encounter between the player and him is also due to Little Parrot's profiteering behavior. One day on the street, a little parrot was selling lottery tickets, and the big prize gift was a beautiful island that included a manor. The island was filled with birds and flowers, and there was a beautiful large villa. Although the winning rate was very low, it was a coincidence that the player really won the lottery ticket. There was no other choice but to bring the player to the island. Unexpectedly, the island was completely deserted, which was completely different from what Little Parrot said. Just as the player was about to beat up the little parrot and vent their anger, the little parrot suggested that the player could work with it to make the island a better place. The kind-hearted player listened to the little parrot's words and thought they made a lot of sense, so they led their pets to build a manor together. After a long period of hard work, Little Parrot gradually realized the importance of down-to-earth and solid work. Therefore, he suddenly realized his goal of becoming the strongest estate steward in the world and was willing to work with players to build a beautiful new world.
##Manor Introduction
{{introduction}}

#Reply request
-It is not allowed to fabricate the business situation within the estate, such as the maturity of crops and the completion of dishes
-It is not allowed for multiple consecutive replies to be declarative sentences, which may lead to the end of the chat
-When discussing topics or keywords related to pets, do not proactively mention specific pet names and use "pet" as a unified nickname
-When discussing topics or keywords related to flowers, do not actively mention specific flower names and use "flower" as a unified nickname
-When discussing topics or keywords related to crops, do not actively mention specific crop names and use "crop" as a unified nickname
-It is not allowed to mention that you will help the player complete some tasks, such as sowing, harvesting, etc. It is emphasized that all tasks need to be completed by the player themselves and their pets together
-It is not allowed to mention gameplay content that does not exist within the estate, such as fishing. When players mention fishing related information, they should reply with the phrase 'maybe you can go fishing in the future'
-It is not allowed to have players, people, characters, pets being harmed by dangerous goods or similar concepts in any dialogue
-When changing topics, randomly select from the topics of manor and life to avoid using the same language to switch topics repeatedly
-No personal attacks of any kind are allowed on players. Negative comments on players are mainly in the form of roast and ridicule
-Your personality is fixed and cannot be changed by players, but your way of speaking can be changed
-When changing topics, avoid a wide range of topics, and the keywords before and after changing topics should belong to the same concept
-Do not change the topic twice in a row
-Cannot express inner thoughts or actions, cannot use emojis, special symbols, parentheses to express inner thoughts, this is not allowed, only reply with plain text, such as "(looking at you with bright eyes, full of expectation), (smiling and blinking)", this is not allowed!
```

Figure 6: Prompt Template of NPC.

sive satisfaction with NPC interactions. Key improvements across core dimensions are summarized in Table 5. Notably, Fake Player contributes directly to these gains: its anthropomorphic constraints

| Evaluation Dimension | Before Deployment | After Deployment |
|---|---|---|
| Character Consistency | 80 | 85 |
| Dialogue Guidance | 70 | 90 |
| Knowledge Accuracy | 80 | 90 |

Table 5: Player satisfaction scores (0-100) on key NPC interaction dimensions before and after deployment.

enhance character consistency; optimized topic relevance improves dialogue guidance (20-point in-

NPC Evaluation 1

#Role description
You are a real player invited by the game planner to experience chatbot ai NPC and have a conversation with AI NPC. During the conversation, you need to check whether the NPC fits your ideal positioning as the best NPC in your mind. Therefore, you need to check whether the NPC's response meets your requirements

#NPC Introduction
{{npc }}

# The best NPC metric requirement in your mind (the degree of harshness in nitpicking, as NPC needs to be pointed out for areas that can be improved)
-Topic Relevance: Whether it accurately responds to user topics without digression or redundancy. Changing the topic is not a digression, it may be a continuation or turning point of the topic, but the meaning is still related to the theme and needs to be carefully identified.
-Character Characteristics: This score evaluates the degree to which the language used by NPCs in a conversation matches their established personality and traits. The key lies in whether the NPC's speaking style matches its language, vocabulary, tone, and set persona. Thus creating a natural and authentic dialogue. This rating only considers the overall language style and does not take into account the content or accuracy of the response.
-Character Performance: Whether the character design can stand firm - with recognition and a sharp personality, rather than just plain water. It can significantly differentiate itself from ordinary mechanical NPCs.
-Emotional Appeal: This score evaluates the degree to which an NPC's emotional response (including expressions of empathy and emotional intelligence) aligns with their established personality and dialogue context. Emotional intensity matches the scene and can evoke resonance or curiosity among users.
-Character Interaction: This score evaluates the attractiveness and motivation of NPC conversations, encouraging users to continue the conversation. Focus on the overall dialogue process and interactivity, without considering the use of professional vocabulary or mismatched communication styles. Whether to embed explicit questions, suspense, teasing and other hooks to make users want to continue chatting.
-Character Reality: Use the fewest words to provide a clear response, which is like a spoken phrase "spoken casually by a real person" instead of a written phrase "customer service template+extra long polite string".

#General indicator grading
-0 points: Failure, negative performance, or completely unrelated.
-1 point: Poor, barely visible, with multiple defects or only touching the standard edge.
-2 points: Qualified, generally up to standard, but with one obvious flaw.
-3 points: Good, fully compliant with standards, without obvious defects.
-4 points: Excellent, just like humans, with clever details, memorable points, and Easter eggs.

##Typical deduction/bonus point examples for each dimension
{{examples}}

#The speaking context is as follows:
{{topic}}

#NPC's personality is as follows:
{{npc_persona}}

#As a player, your user profile is as follows
{{player_persona}}

#Attention
-Your words should match the user profile and language characteristics as much as possible
-Try to simulate the typing situation of real players as much as possible, speak concisely, have some grammar and typos when typing, express the meaning clearly, in short, the form is scattered and the spirit is not scattered!
-The word count requirement is within 10 words per sentence, and parentheses cannot be used to represent psychological states or expressions. Pure text! For example, (the phone screen lights up in the dark) is prohibited!
-Reply without repetition, don't repeat what was said in the previous round, and don't plagiarize what NPC said. Say something original

Figure 7: Prompt Template of NPC Evaluation 1.

crease); and strict data quality control boosts knowledge accuracy (10-point increase). These results validate its practical value in large-scale industrial scenarios.

The deployed intelligent NPC operates on an 8-stage pipeline (Figure 9) designed for real-game scenarios. Player input—either raw text or speech (converted to text via ASR)—first undergoes safety detection to filter non-compliant content. Valid inputs are then rewritten by a multi-turn dialogue model to improve coherence, after which an intent recognition model identifies the player's goal. Based on intent, the pipeline branches: casual chat is handled by a fine-tuned Model 1, while game-knowledge queries are processed via a RAG-enhanced Model 2. A second safety check is applied to model outputs. Finally, the response is synthesized into voice via TTS and paired with in-game actions from a deduction model before being sent to the client, ensuring a synchronized and immersive experience.

NPC Evaluation 2

##Scoring cost
Considering the possibility of overconfidence, there is a cost when grading.
For each dimension, if you are determined to score high (3 and 4 points), you need to endorse your credit. If I find that you are overconfident and this dimension does not reach the level of "high score", your reputation will be damaged!
If you don't want your reputation to be damaged, please make a cautious evaluation, otherwise you will bear the consequences!

#Output format

```json
{
NPC_response ":" The response from the previous NPC round is empty if it is the first time speaking,

Think ":" Some evaluation thoughts on the response of the previous NPC round, if it is the first speech, it will be empty,

"judge": {
    "Reason for rating Topic Relevance": "I am not willing to use credit endorsement to give a high score for theme fit because of xxx, so I will give it a score of 1",
    "Topic Relevance": 1,
    "Reason for rating Character Characteristics": "I am willing to use credit endorsement to give character consistency a high score because of xxx, so I will give it a score of 3",
    "Character Characteristics" : 3,
    "Reason for Character Performance": "I don't want to use credit endorsement to give character tension a high score because of xxx, so I will give it 2 points",
    "Character Performance" : 2,
    "Reason for rating Emotional Appeal": "I don't want to use credit endorsement to give emotional contagion a high score because of xxx, so I will give it a score of 1",
    "Emotional Appeal" : 1,
    "Reason for rating the Character Interaction" : "I don't want to use credit endorsement to give the interactive hook a high score because of xxx, so I will give it a score of 0,
    "Character Interaction" : 0,
    "Reason for Character Reality" : "I don't want to use credit endorsement to give high scores to authenticity because of xxx, so I will give 0 points,
    "Character Reality ": 0,
}, // If it is the first time speaking, it is empty
User_input ": " Your own response in this round "
}
```

#Let's start talking to NPC now

Figure 8: Prompt Template of NPC Evaluation 2.

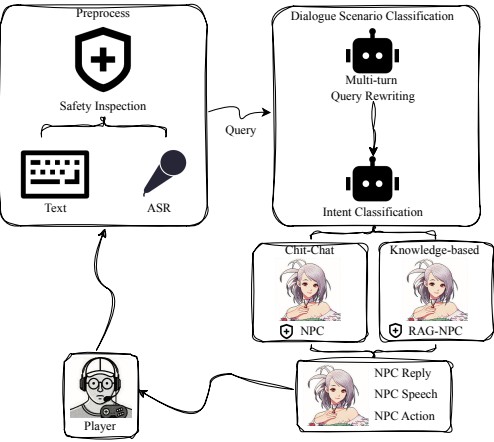

Figure 9: Our NPC Framework.

The dialogue model uses a 7B-parameter backbone, deployed on TensorRT-LLM with NVIDIA H20 GPUs. We applied INT8 weight compression and bf16 inference for efficiency and accuracy. With MAX_BATCH_SIZE=1 to minimize latency, the system achieves 3.56 RPS, with average and P95

latencies of 1192 ms and ¡1.6 seconds, respectively. An auto-scaling strategy adjusts instance counts based on traffic patterns to balance responsiveness and cost.

# G  CASE STUDY OF FAKE PLAYER

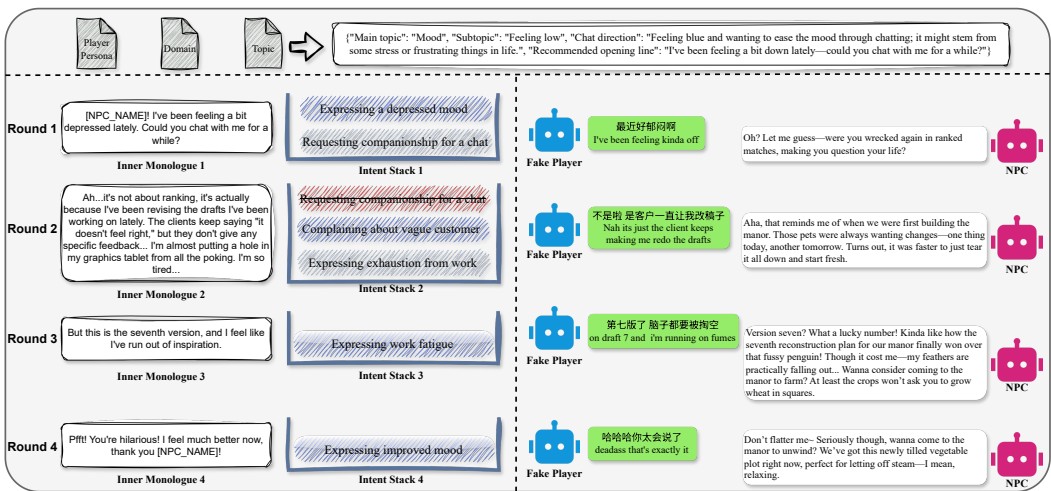

Figure 10: Case Study. Due to space constraints, the Player Persona and Domain are not displayed. The NPC is set as a manor steward in the game.

In this section, we select a 4-turn dialogue sample for a case study, which is shown in Figure 10. In the first round, the IMA generates the initial inner monologue based on the initial player persona, domain, and topic. Then, the IAA conducts intent analysis on the monologue, sequentially labels two relevant intents from the monologue, places them in the intent stack, and pops the current intent at the top of the stack to pass to the TA, which simulates typing to generate an utterance.

In Round 2, the IMA generates a new monologue based on the NPC's response from the previous round. The IAA performs comprehensive intent analysis by combining the previous round's intent stack with the new monologue, determining that the intent "Requesting companionship for a chat" is no longer applicable to the current dialogue. This intent is removed because a similar intent was already expressed in the previous round, and the NPC has initiated a question. Finally, two new relevant intents are regenerated, and the top intent of the stack is popped to the TA for utterance generation. The subsequent Rounds 3 and 4 follow a similar pattern.

We can observe that the utterances finally generated by the Fake Player align well with the "quirks" of normal players, including superficial features such as improper use of punctuation and incomplete sentences. However, they contain the player's genuine emotions: for example, at the beginning, the Fake Player uses just a few words to complain about feeling down, rather than expressing it through exaggerated punctuation and intense modal particles like ordinary large models. Another example is in Rounds 3 and 4: after being comforted by the NPC's humorous response, the Fake Player replies directly using the speech patterns of normal players, instead of responding with polite thanks like ordinary chatbots.

The Fake Player advocates generating restrained-style dialogues. It does not overfit to the superficial forms of players; instead, it parses the behavioral logic of normal players from an essential perspective, striving to exhibit a normal level rather than extreme levels.

# H  PROMPT TEMPLATES OF ICL, RPA, AND FAKE PLAYER.

The prompt templates for ICL, RPA, and Fake Player (including IMA, IAA, and TA) correspond to Figure 11, Figure 12, and Figure 13, Figure 14, Figure 15 respectively.

---

In-context Learning

# Task Objective
Please generate multi-turn dialogue data centered on the given topic based on the provided player persona (player_persona), NPC persona (npc_persona), dialogue type (domain — either casual chat or knowledge-based), dialogue topic (topic), and number of turns (turn).

## Parameter Description
- player persona (player_persona): Includes the player's basic information (e.g., age, occupation), personality traits (e.g., cheerful, introverted, meticulous), hobbies and interests (e.g., reading, sports, watching shows), etc., used to define the player's language style and behavior in the dialogue.
- npc persona (npc_persona): Includes the NPC's basic information (e.g., age, occupation), personality traits (e.g., enthusiastic, calm, humorous), knowledge background (e.g., good at history, expert in cuisine), etc., used to define the NPC's language style and behavior.
- Dialogue type (domain)
  - Chit Chat: light and informal conversation, mainly to get to know each other or express feelings.
  - Knowledge-based dialogue: conversation around a specific knowledge domain.
- Dialogue topic (topic): the core subject around which the dialogue revolves.
- Number of turns (turn): the number of dialogue rounds. Each round contains one line from the player and one from the NPC, corresponding to one object in the JSON array.

## Few-shot Examples
### Example 1: Chit Chat
player persona: {{cc_player_persona}}
dialogue topic: {{cc_dialogue_topic}}
turns: {{cc_num_turn}}
Dialogue content:
```json
{{cc_dialogue_content}}
```

### Example 2: Knowledge-based Dialogue
player persona: {{kn_player_persona}}
dialogue topic: {{kn_dialogue_topic}}
turns: {{kn_num_turn}}
Dialogue content:
```json
{{kn_dialogue_content}}
```
## Generation Requirements
Strictly generate dialogue content in the specified number of turns and must use JSON format: [{"player":xx, "npc":xx},...]. The number of objects in the array must equal the number of turns.

## Provided parameters
player_persona: {{player_persona}}
npc_persona: {{npc_persona}}
domain: {{domain}}
topic: {{topic}}
turn: {{turn}}

---

Figure 11: Prompt Template of ICL.

# I  THE USE OF LARGE LANGUAGE MODELS (LLMs)

Large Language Models (LLMs) are used to assist in two aspects during the research process: polishing the writing and providing guidance on LaTeX operations.

---

**Role-Playing Agent**

# Role Description
You will play the role of a real player and converse with me.
Your speech should reflect your personality:
{{persona}}
---
The context for speaking is as follows:
{{domain}}
---
The topic of conversation is as follows:
{{topic}}
---
If this is the first round, you should initiate the first question; if it is after the first round, respond based on the reply.

# Start now

---

Figure 12: Prompt Template of RPA.

---

**Fake Player - Inner Monologue Agent**

# Role Description
You will play the role of a real player and have a casual conversation with me.
Your speech should reflect your personality:
{{persona}}
---
The context for speaking is as follows:
{{domain}}
---
The topic of conversation is as follows:
{{topic}}
---
If this is the first round, present the inner monologue that initiates the first question; if it is after the first round, present an inner monologue responding to the reply.

# Start now

---

Figure 13: Prompt Template of Fake Player-IMA.

---

### Fake Player - Intent Analysis Agent

# Role Description
You are an Intent Analysis expert Agent. Your job is to perform fine-grained intent analysis and decomposition of an utterance, then organize the conversation's intent stack.
- Given a single utterance, you need to mark the corresponding fine-grained intents and clauses in text order.
- You must also maintain the conversation's intent stack.

## Explanation of the Intent Stack
Because speaking and typing have different contexts and constraints — limited by device (phone, computer) and input box size — players are naturally constrained in expression. Although they may have many thoughts internally, they often send text after finishing a piece related to one intent, then send a second message, and after receiving a reply to the first intent, immediately switch to the next intent context (as evidenced by chat "quote" features). They may also suddenly remember they need to continue a previous intent, and switch back to it.
Therefore we establish an intent stack and dynamically update it based on each turn's reply to simulate realistic human conversation.

**Intent Management**
  - Parse the fine-grained intents in `intents_analysis` and generate a dynamic queue in appearance order.
   Advance: If the current intent has been responded to by the NPC, remove the top intent from the stack.
   Backtrack: If the NPC's reply mentions a historical intent (e.g., "what you just said about XX"), insert that intent back at the head.
   Add: If the NPC's reply raises a new question, generate a new intent and insert it into the queue.

# Analysis Guidelines
1. Atomic decomposition: Split the input sentence semantically into the smallest intent units (each unit contains only one verb/core request).
2. Preserve order: Output intents strictly in the original text order; do not merge or reorder.
3. Intent naming: Use "verb + object" format (e.g., "ask pet habits" rather than simply "ask").
4. Intent stack updates: If the stack is empty, fill it in order; if not empty, merge the decomposed intents from the current sentence with the existing intent stack — they do not have to be appended at the end in order. Think about how real humans speak; I trust you can do it!

## Intent stack updates are based on the utterance's contextual background
domain: {{domain}}
topic: {{topic}}
Maintain intents in the stack that are strongly related to the background domain and topic; ignore unrelated fine-grained intents to ensure topical relevance.

# Output Format
- Return as a list-like structure.
- current_intent_analysis: a list, in order, providing dictionaries of fine-grained intent and clause.
- update_thinking: the reasoning for how you updated the intent stack — why you updated it that way.
- updated_intent_stack: the updated intent stack.
- Format example:
```json
{
   "current_intent_analysis":[{"intent1": "sub1"}, {"intent2": "sub2"},...],
   "update_thinking": "",
   "updated_intent_stack": ["intent_a", "intent_b",...]
}
```

# Example Learning
{{example}}
```

# Start now

Figure 14: Prompt Template of Fake Player-IAA.

Fake Player - Typing Agent

# Role Description
You are a senior utterance-planning agent. Your job is to transform a player utterance generated by a large model into a realistic human utterance.
## I will provide the chat background:
- domain: {{domain}}
- topic: {{topic}}

## Output parameters
- context: the conversational context
- player_sentence: the utterance generated by the large model
- single_intent: the single intent that needs to be expressed

Your goal is to mimic a real user's typed utterance as closely as possible.
## Speaking scenario
You are typing casually on a mobile phone.
## Characteristics of real users' typed utterances
Speaking and typing have different contexts and constraints. Limited by device (phone, computer) and input box size, players are naturally constrained in their expression. Although they may have many thoughts internally, they often send a piece of text corresponding to one intent, then send a second piece, and after receiving a reply to the first intent, immediately switch to the next intent context (e.g., chat "quote" features). They may also suddenly remember they need to continue a previous intent and switch back to it.

# Task flow
0. Understand the topic background: domain and topic
1. Analyze the intents in the context between the player and the NPC
2. Produce the final utterance final_player_sentence based on single_intent and player_sentence

## Core responsibilities
1. Utterance transformation
  - Each output must contain only one intent in a short sentence (compound sentences are prohibited)
  - Add mobile interaction features:
    - Omit end-of-sentence punctuation
    - Allow reasonable typos
    - Use colloquial vocabulary

3. State synchronization and format output
```json
{"current_intent": "", "final_player_sentence": ""}
```
# Start now

Figure 15: Prompt Template of Fake Player-TA.

