# OpenReview forum: "Fake Player: Imitating Real Player to Distill Data for LLM-based NPC Training"
_ICLR.cc/2026/Conference — Submitted to ICLR 2026_

### Official Review · Reviewer_NmYs · 2025-10-29

**Soundness:** 3
**Presentation:** 3
**Contribution:** 3
**Rating:** 6
**Confidence:** 3

**Summary:**

This paper addresses a critical gap in the development of LLM-based NPCs: the lack of diverse and authentic player-side data for fine-tuning. The authors argue that existing methods focus too heavily on NPC persona-alignment while neglecting the quality of the player inputs, leading to models that fail in real-world interactions.

To solve this, they propose "Fake Player," a novel multi-agent distillation framework. This framework simulates a human player by decomposing the interaction process into three stages: an Inner Monologue Agent (generating internal thoughts), an Intent Analysis Agent (extracting and stacking fine-grained intents), and a Typing Agent (converting the top intent into a concise and human-like utterance). This approach aims to generate data that is both diverse and behaviorally realistic.

Additionally, the paper introduces "Distill Bench", a benchmark for quantitatively assessing the quality of synthetic dialogue data before its use in training. Experiments show that Fake Player generates superior data compared to baseline methods (ICL and Role-Playing Agents) and that NPCs fine-tuned on this data significantly outperform others, even allowing smaller 7B models to compete with larger prompt-based ones.

**Strengths:**

1. The paper clearly identifies and tackles a practical, high-impact problem. The insight that the authenticity of player-side data is a bottleneck for fine-tuning robust NPCs is both non-obvious and significant for the field.

2. The "Fake Player" architecture is a good solution. The conceptual model of a player having complex "inner monologues" but being limited by a "typing constraint" to express one intent at a time is an effective way to model human-NPC interaction in games. This principled approach is a clear improvement over more naive agent-based or few-shot generation methods.

3. The authors prove the effectiveness of the synthetic data by fine-tuning models and measuring their performance. The proposal of a standardized benchmark to evaluate synthetic data before training is also a valuable secondary contribution. This provides a necessary tool for researchers and developers, saving significant time and computational resources that would otherwise be spent on iterative fine-tuning and evaluation cycles.

**Weaknesses:**

1. The three-agent pipeline, while effective, is inherently more complex than the baselines. This paper would benefit from a brief discussion of the computational cost and latency of generating data using Fake Player versus the simpler ICL and RPA methods. It's important to understand the trade-offs between data quality and data generation efficiency.

2. It is unclear how this "typing agent" work. Moreover, it is recommended to supplement the case study of "typing agent" (e.g., the prompting template and the synthetic human output) for a more comprehensive understanding. Moreover, The "typing agent" is noted to introduce "stochastic imperfections" like typos and grammatical errors. While this intuitively adds to "human-likeness," the paper does not present an ablation study on how much this specific feature actually contributes to the downstream model's robustness or performance. It would be valuable to know if this is a critical component or a minor addition.

3. For evaluation, some other NPC creation/style adaptation methods could also be considered besides fine-tuning, such as representation editing (e.g., Ma et al. 2025, DRESSing up LLMs, etc.). It could be more persuasive if the dataset is useful for broader types of methods.

4. Regarding the Distill Bench evaluation (Table 1), the "Human-Likeness" score for Fake Player (e.g., 69.80) is a significant improvement over baselines but still notably lower than the score for real players (80.20, from Table 3). This gap is more significant compared to that between different methods. What do you believe accounts for this remaining gap, and what future work could help close it?

**Questions:**

See Weaknesses.

---

> ### Author Response · Authors · 2025-11-21
> **(1) Response to Reviewer NmYs**
>
> Thank you for reviewing our work. Below, we will address your concerns and suggestions point by point:
>
> # 1 About the computational cost
> We quantified the cost and latency of Fake Player vs. baselines using DeepSeek-V3-250324 (cost: 2 CNY/1M input tokens, 8 CNY/1M output tokens; 1 USD ≈ 7.14 CNY). For 500 multi-turn samples (concurrency=50):
>
> | Method | Input Token | Input Cost (CNY) | Output Token | Output Cost (CNY) | Total Cost (CNY) | Total Cost (US Dollar) | Time Consumption (s) |
> | --- | --- | --- | --- | --- | --- | --- | --- |
> | ICL | 4106549 | 8.21 | 246121 | 1.97 | 10.18 | 1.43 | 100 |
> | RPA | 5834138 | 11.67 | 324212 | 2.59| 14.26| 1.99 | 402 |
> | Fake Player | 6231786 | 12.46 | 574966 | 4.60 | 17.06 | 2.39 | 596 |
>
> For reference, manually annotated multi-turn dialogue data costs an average of 5 CNY per sample—generating 500 samples would cost 2500 CNY (≈ 350 US Dollars), which is far higher than the costs of all three automated methods.
>
> While Fake Player’s synthesis cost is slightly higher than that of ICL and RPA, it still remains in a highly cost-effective range (and is vastly cheaper than manual annotation). More importantly, ICL and RPA fail to meet the quality requirements for product deployment due to issues like insufficient diversity or poor human-likeness; in contrast, **Fake Player achieves production-ready data quality** with its balanced performance in authenticity, diversity, and topic relevance. This high-quality data further helps save downstream NPC model training costs (e.g., reducing iterations caused by low-quality data).
>
> Notably, for in-development games planning to deploy large-scale NPCs, Fake Player’s cost-effectiveness will be even more prominent: its ability to generate a large volume of high-quality data from minimal seed settings avoids the exponential labor and time costs of manual annotation or repeated prompt polishing for other methods.
>
> # 2 About the Typing Agent
> Thank you for your suggestion. We will add a corresponding case study to the manuscript in subsequent revisions. Additionally, the design details of Fake Player (including the Typing Agent) have been provided in the supplementary material—we apologize for not explicitly indicating this in the main text and will supplement this reference in the manuscript to enhance clarity. We also plan to open-source our method and the baselines on GitHub; as this is a method that has been successfully deployed in industrial scenarios, we hope it can contribute to the research community.
>
> The "random imperfections" (e.g., typos) simulated by the Typing Agent are rooted in targeted modeling of Chinese players. In Chinese game interaction scenarios, players typically use Pinyin input methods on mobile devices or keyboards, which inherently lead to a high probability of typos. To align with this real-user behavior, the Typing Agent is designed to simulate such common typo scenarios, ensuring the synthesized data matches actual player typing habits.
>
> Notably, this feature serves as a "value-added optimization" rather than a core functional component. Thus, it does not introduce any adverse effects on the downstream performance of NPC models. We appreciate your attention to this detail, and this feedback will inspire us to explore more refined human behavior modeling in future research.
>
> # 3 About the extension to representation editing
> Thank you for your suggestion on this insightful direction. The core contributions of this paper lie in developing a high-quality synthetic data generation pipeline (Fake Player) and a standardized evaluation benchmark (Distill Bench). To clearly demonstrate the core advantages of our method, we prioritized validating the effectiveness of the distilled data through fine-tuning—a widely adopted approach in the LLM-based NPC field. This choice ensures that the value of our data (in terms of authenticity, diversity, and topic relevance) is directly and unambiguously reflected in downstream NPC performance.
>
> We fully agree that extending the evaluation to representation editing and other style adaptation methods would further highlight the generality of our distilled data. This is indeed a promising future research direction: the high-quality, human-aligned data generated by Fake Player has the potential to support diverse NPC creation paradigms beyond fine-tuning. We plan to systematically explore this direction in subsequent work, aiming to comprehensively assess the application value of our data across more NPC development scenarios and provide richer insights for the community.

---

> > ### Author Response · Authors · 2025-11-21
> > **(2) Response to Reviewer NmYs**
> >
> > # 4 About the Human-Likeness experiment
> > Our results show that Fake Player achieves a significant improvement in Human-Likeness compared to baseline methods (ICL and RPA), yet its score remains notably lower than that of real human players.
> >
> > The primary reason for this gap lies in the general-purpose LLM we used for data generation: the training corpus of such models is biased toward formal written language, which inevitably leads to a slight "written-style residue" in the distilled data—making it difficult to fully match the casual, colloquial characteristics of real player interactions (e.g., fragmented expressions, conversational slang).
> >
> > To address this, we plan to refine the data generation process in future work: we will clean and extract real player queries from user interaction logs to train a task-specific user simulator, which is expected to further enhance the Human-Likeness of synthesized data by aligning more closely with authentic player language habits. Currently, we have initiated relevant research and intend to open-source the weights of the trained user simulator at an appropriate time.
> >
> > **Thank you for your valuable suggestion—it will undoubtedly help improve the quality of our work!**

---

> > > ### Comment · Reviewer_NmYs · 2025-11-27
> > > **Acknowledgements of rebuttal**
> > >
> > > I thank the authors for their response and efforts to the manuscript. I have no further questions and am happy to maintain my original positive score.

---

> > > > ### Author Response · Authors · 2025-11-28
> > > >
> > > > Dear Reviewer NmYs
> > > >
> > > > Thank you for recognizing the contributions of our work. We also appreciate your insightful suggestions, which inspired us and helped improve the quality of our paper.
> > > >
> > > > best wishes,
> > > > Authors of Paper 17228

---

### Official Review · Reviewer_PUtu · 2025-10-31

**Soundness:** 2
**Presentation:** 2
**Contribution:** 2
**Rating:** 6
**Confidence:** 4

**Summary:**

This paper targets the critical gap in training data for LLM-based game NPCs: the lack of diverse, high-fidelity player-side inputs. The authors argue that existing data generation methods like In-Context Learning (ICL) or Role-Playing Agents (RPA) produce data that is either low-diversity or not human-like.

To solve this, the paper introduces two key contributions:
1.  **`Fake Player`**: A novel multi-agent distillation framework designed to simulate a human player's cognitive and physical interaction constraints. It consists of an `Inner Monologue Agent` (to generate thoughts), an `Intent Analysis Agent` (to manage a stack of conversational intents), and a `Typing Agent` (to convert the top intent into a concise, device-constrained, human-like utterance).
2.  **`Distill Bench`**: A new benchmark for quantitatively assessing the quality of the *distilled data itself* on dimensions of Topic Relevance, Human-Likeness, and Data Diversity, bypassing the need for costly downstream NPC retraining.

Experiments demonstrate that `Fake Player` generates higher-quality data than baselines and that small 7B-parameter models fine-tuned on this data can achieve performance approaching that of large, prompt-based models like GPT-5.

**Strengths:**

The paper's primary strength lies in its novel and well-motivated `Fake Player` framework. Instead of just generating "player" text, it models the *underlying process* of human interaction, separating internal monologue from a device-constrained typed utterance via an intent stack. This is a significant and original idea. The second major strength is the `Distill Bench`, which provides a much-needed, low-cost method for evaluating the quality of synthetic training data. The experimental results are strong, particularly the downstream task analysis showing a fine-tuned 7B model approaching GPT-5's performance, and the industrial deployment confirms real-world significance.

**Weaknesses:**

1.  **Missing Ablation Study:** The `Fake Player` framework is presented as a three-component system: `Inner Monologue Agent` (IMA), `Intent Analysis Agent` (IAA), and `Typing Agent` (TA). While the full system shows strong results, the paper lacks an ablation study to justify this specific architecture. It is unclear what the relative contribution of each agent is. For instance, how much of the "Human-Likeness" gain comes from the `Typing Agent`'s conciseness constraints versus the `Intent Analysis Agent`'s sophisticated intent stack? Would a simpler model (e.g., IMA -> TA) perform similarly? This is a key omission for understanding *why* the method works so well.

2.  **Over-reliance on LLM-as-Judge:** The evaluation, both for `Distill Bench` and the downstream NPC task, heavily relies on "LLM as a Judge" (specifically, DeepSeek-V3). While this is a common practice, it introduces potential biases. The judge's high scores for `Fake Player`'s output might stem from a shared stylistic bias (as `Fake Player` also uses LLMs) rather than true human-likeness. The human evaluation in Table 3 is a good inclusion to mitigate this, but it is very small (only 35 dialogues) and only evaluates the *data*, not the final *NPC interaction*.

3.  **Lack of Cost/Efficiency Analysis:** The `Fake Player` framework appears significantly more computationally expensive for data generation than the baselines. Algorithm 1 implies that each turn of generated dialogue requires multiple sequential LLM calls (IMA, IAA, TA). This is a 3x or more increase in generation cost compared to ICL (one call) or RPA (one call per agent, often parallelizable). The paper does not discuss this trade-off. For a method focused on distillation, the *cost of distillation* is a critical, practical factor that has been overlooked.

4.  **Limited Scope of Evaluation:** The downstream evaluation is based on a single NPC persona (an "estate manager") in what appears to be a "companionship" context. It is not clear how well the `Fake Player`'s assumptions (e.g., concise, typo-prone inputs) would generalize to other game genres or interaction types, such as an "antagonist" NPC, a "quest-giver" NPC requiring complex informational queries, or a fast-paced action-game helper.

**Questions:**

1.  **On Validating the Judge:** The reliance on "LLM as a Judge" is a potential weakness. While Table 3 provides a small human comparison for "Human-Likeness", could the authors provide more detail on the validation of the DeepSeek-V3 judge? For example, was any inter-rater reliability (e.g., Cohen's Kappa) measured between the LLM-judge and human evaluators on a larger subset of `Distill Bench` to ensure the judge's ratings for all dimensions (especially "Topic Relevance" and "Improvisation") are truly aligned with human preferences?

2.  **On the Necessity of the Intent Stack:** Given the lack of an ablation study, could the authors provide insight into the specific contribution of the `Intent Analysis Agent`? What measurable drop in data quality (e.g., in "Human-Likeness" or "Topic Relevance" scores) would be expected if this agent were removed, and the `Typing Agent` was prompted to simply "convert the inner monologue into a concise, human-like utterance"?

3.  **On Data Generation Costs:** Could the authors elaborate on the computational cost (e.g., API calls, tokens processed, or wall-clock time) of generating the training dataset using `Fake Player` versus the ICL and RPA baselines? How does this cost factor into the overall practical utility of the method, especially for smaller developers?

4.  **On Generalizing Interaction Styles:** The `Typing Agent` simulates conciseness and "stochastic imperfections" typical of casual, "expression-constrained" interactions. How does this framework adapt to the "knowledge-seeking" domain, where a real player might realistically input a much longer, more complex, and grammatically correct query to get a specific answer? Does the `Typing Agent`'s behavior change based on the `Domain $D$` parameter?

5.  **On Controlling Imperfections:** How were the "stochastic imperfections" (typos, grammatical errors) of the `Typing Agent` implemented? Was this achieved via prompting, or through a separate simulation/post-processing step? Furthermore, how sensitive was the final NPC performance to the *rate* of these imperfections? Is there a "sweet spot," and does too much "realism" (i.e., too many typos) start to degrade the NPC's ability to understand the player?

---

> ### Author Response · Authors · 2025-11-21
> **(1) Response to Reviewer PUtu**
>
> Thank you for reviewing our work. Below, we will address your concerns and suggestions point by point:
> # 1 On Validating the Judge
> Our original intention in designing Distill Bench is to establish an efficient automated evaluation process to support the rapid iteration of data distillation evaluation training. Therefore, we use LLM as judge as the core evaluation method, which can significantly reduce the dependence on manual annotation.
>
> In particular, the selection of this assessment method is based on rigorous preliminary verification. At the beginning of the study, we have systematically verified the effectiveness of the LLM as a judge evaluation criteria adopted in this paper through the manual evaluation of a large number of team members (including the author of the paper). After confirming that it is highly consistent with the results of manual evaluation, the iterative optimization of subsequent models establishes an efficient model based on "LLM as a judge, supplemented by a small amount of manual experience verification".
>
> In order to further prove the relevance of LLM evaluation and human evaluation, we designed a more extensive validation experiment: 50 multi-turn dialogue samples were randomly selected from each method (ICL, RPA, Fake Player), and five independent evaluators outside the team (all with graduate degrees, majors covering computer science and psychology, and in-depth understanding of the game field) were employed to score manually according to the criteria completely consistent with LLM evaluation. The experimental results are shown in the table below:
>
>
> | **Methods**   | **Topic Breadth** | **Topic Depth** | **Total (Topic Relevance)** | **Conciseness** | **Improvisation** | **Total (Human-Likeness)** |
> |---------------|-------------------|-----------------|------------------------------|-----------------|-------------------|-----------------------------|
> | ICL           | 66.2              | 72.2            | 69.15                        | 71.2            | 66                | 68.6                        |
> | RPA           | 72.4              | 67.6            | 70                           | 40.4            | 57.2              | 48.8                        |
> | Fake Player   | 75.2              | 70.4            | **72.8**                         | 74.4            | 72.4              | **73.4**                        |
>
> The analysis shows that the evaluation results of external personnel are consistent with the team's previous verification conclusion: the manual evaluation has a stronger ability to identify the sentences obviously generated by AI, which leads to the humanoid score of RPA method is significantly lower than that of LLM evaluation. However, in the key dimensions, the manual evaluation and LLM evaluation show a high degree of consistency: in terms of topic relevance, fake player performs best; In terms of pseudo humanity, the order of the three is FP>ICL>RPA. This result is completely consistent with the evaluation trend of using DeepSeek-V3 as the judge model.
>
> #  2 On the Necessity of the Intent Stack
> The Intent Stack, as a core component of the Intent Analysis Agent (IAA), is designed to simulate the intent transition logic of real human players during conversations—an insight derived from analyzing complex interaction data in industrial scenarios. Real player dialogues are not random: for example, a player might start by asking about "manor crop planting" (Intent 1), then shift to "pet feeding tips" (Intent 2) after the NPC mentions pets, and later return to "crop pest control" (Intent 1) to follow up on the initial topic. This dynamic, context-aware intent switching (with prior intents temporarily "stored" and reactivated) is a key trait of human-like interactions—one that naive agent models often fail to capture.
>
> To validate its necessity, we conducted an ablation experiment where the IAA (and thus the Intent Stack) was removed. As shown in the table below, the absence of the IAA led to a 2.34-point drop in Human-Likeness (from 69.8 to 67.46) in the Distill Bench evaluation, while other dimensions (e.g., Topic Relevance, Diversity) showed only minor fluctuations. This directly confirms that the Intent Stack is critical to modeling real player behavior—its ability to track, update, and prioritize intents ensures the synthesized dialogue reflects the natural, non-linear intent transitions of human players.
>
> | **Methods** | **Topic Breadth** | **Topic Depth** | **Total (Topic Relevance)** | **Conciseness** | **Improvisation** | **Total (Human-Likeness)** | **Diversity**| **Overall** |
> | --- | --- | --- | --- | --- | --- | --- | :---: | :---: |
> | Fake Player | 73.2 | 75 | 74.1 | 64.40 | 75.20 | 69.8 | 67.28 | 70.39 |
> | w/o IAA | 73.7 | 74.5 | 73.9 | 62.56 | 72.36 | 67.46 | 66.41 | 69.26 |

---

> > ### Author Response · Authors · 2025-11-21
> > **(2) Response to Reviewer PUtu**
> >
> > In summary, the Intent Stack is not a redundant design—it addresses the complexity of real player intent dynamics observed in industrial data, directly boosting the human-likeness of synthesized interactions and laying the foundation for high-quality NPC training data.
> >
> > # 3 Regarding Data Generation Costs
> >
> > We quantified the cost and latency of Fake Player vs. baselines using DeepSeek-V3-250324 (cost: 2 CNY/1M input tokens, 8 CNY/1M output tokens; 1 USD ≈ 7.14 CNY). For 500 multi-turn samples (concurrency=50):
> > | Method | Input Token | Input Cost (CNY) | Output Token | Output Cost (CNY) | Total Cost (CNY) | Total Cost (US Dollar) | Time Consumption (s) |
> > | --- | --- | --- | --- | --- | --- | --- | --- |
> > | ICL | 4106549 | 8.21 | 246121 | 1.97 | 10.18 | 1.43 | 100 |
> > | RPA | 5834138 | 11.67 | 324212 | 2.59| 14.26| 1.99 | 402 |
> > | Fake Player | 6231786 | 12.46 | 574966 | 4.60 | 17.06 | 2.39 | 596 |
> >
> > For reference, manually annotated multi-turn dialogue data costs an average of 5 CNY per sample—generating 500 samples would cost 2500 CNY (≈ 350 US Dollars), which is far higher than the costs of all three automated methods.
> >
> > While Fake Player’s synthesis cost is slightly higher than that of ICL and RPA, it still remains in a highly cost-effective range (and is vastly cheaper than manual annotation). More importantly, ICL and RPA fail to meet the quality requirements for product deployment due to issues like insufficient diversity or poor human-likeness; in contrast, **Fake Player achieves production-ready data quality** with its balanced performance in authenticity, diversity, and topic relevance. This high-quality data further helps save downstream NPC model training costs (e.g., reducing iterations caused by low-quality data).
> >
> > Notably, for in-development games planning to deploy large-scale NPCs, Fake Player’s cost-effectiveness will be even more prominent: its ability to generate a large volume of high-quality data from minimal seed settings avoids the exponential labor and time costs of manual annotation or repeated prompt polishing for other methods.
> >
> > # 4 On Generalizing Interaction Styles
> > The modeling of our Typing Agent is rooted in in-depth analysis of real player dialogue data from game scenarios. Through analyzing interactions in game NPC companionship tasks, we identified two universal traits across both casual chat and knowledge-seeking scenarios:
> >
> > - Conciseness: Players tend to use concise expressions constrained by device interfaces (e.g., mobile keyboards) and conversational efficiency—avoiding overly lengthy or verbose text;
> >
> > - Random imperfections: Due to Pinyin input methods (in Chinese contexts) or quick typing habits, players often produce minor typos, irregular punctuation, or incomplete grammar (e.g., mistyping "副本 (fù běn, dungeon)" as "复本 (fù běn)" or omitting commas in long sentences).
> >
> > These traits are embedded in the Typing Agent’s system prompt to ensure it aligns with real player interaction styles by default. Additionally, as you noted, the Typing Agent is designed with a reserved Domain slot: it adjusts its expression characteristics based on the specific domain (e.g., slightly more formal conciseness for knowledge-seeking dialogues about manor crop cultivation, and more relaxed random imperfections for casual chats about daily pet care). This design allows the agent to generalize across different NPC interaction scenarios while retaining human-like authenticity.
> >
> > # 5 On Controlling Imperfections
> > The simulation of typing imperfections (e.g., typos) is implemented through prompt design.
> >
> > Our motivation for this feature stems from targeted modeling of Chinese players: in Chinese game interaction scenarios, players commonly use Pinyin input methods on mobile devices or keyboards, which inherently leads to a high probability of typos. To align with this real-user behavior, we embedded specific instructions in the Typing Agent’s prompt—guiding it to simulate common typos that players make—thus aligning the synthesized data with actual player typing habits.
> >
> > Notably, this function serves as a "value-added optimization" rather than a core functional component. As such, it does not introduce any adverse effects on the downstream performance of NPC models. We appreciate your attention to this detail, and this feedback will inspire us to explore more refined human behavior modeling in future research.
> >
> > **Thanks for your review again! We hope our responses can address your concerns.**

---

### Official Review · Reviewer_CQsN · 2025-10-31

**Soundness:** 2
**Presentation:** 3
**Contribution:** 2
**Rating:** 4
**Confidence:** 4

**Summary:**

This paper addresses a critical and often-overlooked problem in the training of LLM-driven game NPCs: the generation of high-quality, realistic "player-side" dialogue data for fine-tuning.
The authors argue that existing research focuses heavily on ensuring the NPC's "persona consistency," while neglecting the diversity and authenticity of the user inputs in the training data. This leads to NPCs that perform poorly or break character (Out of Character, OOC) when faced with real-world player inputs, which are often concise and intent-driven.
To solve this, the paper presents two main contributions:
1. Fake Player: A multi-agent LLM distillation framework. This framework uses three collaborating agents (Inner Monologue Agent to generate thoughts, Intent Analysis Agent to extract and maintain an intention stack, and Typing Agent to simulate physical constraints by converting intent to concise text) to mimic constrained, human-like players. This allows for the distillation of diverse and behaviorally-aligned dialogue data from large LLMs.
2. Distill Bench: A standardized benchmark for quantitatively assessing the quality of distilled data. It aims to evaluate data across three dimensions—Topic Relevance, Human-Likeness, and Data Diversity—without requiring costly NPC retraining, thus reducing iterative development costs.
Experimental results show that data generated by "Fake Player" scores higher on "Distill Bench" compared to two baselines (ICL and Role-Playing Agent, RPA). More importantly, in downstream tasks, small models (e.g., Qwen-7B) fine-tuned on "Fake Player" data significantly outperform those trained on baseline data and even approach or exceed the performance of large, prompt-based NPC models.

**Strengths:**

1. Clear and Important Problem Definition: The paper accurately identifies a key bottleneck in the LLM-NPC domain—the quality of player-side training data. As shown in Figure 1, ensuring NPCs can handle realistic, varied player inputs is crucial for player immersion and the practical deployment of small-parameter LLMs. This is a practical and valuable research direction.
2. Innovative Data Generation Framework: The "Fake Player" three-agent architecture is a novel design. It goes beyond simple "role-playing" and attempts to model the human "cognition-to-expression" pipeline. The use of an "intention stack" in the "Intent Analysis Agent" and the simulation of physical interface constraints (leading to concise input) by the "Typing Agent" are insightful and reasonable models for why LLM agent outputs often don't feel human.
3. Practical Evaluation Benchmark (Distill Bench): Given the high cost of NPC training (especially SFT), the proposal of "Distill Bench" has significant engineering value. It provides an "offline" method for evaluating data quality, allowing researchers to rapidly iterate on and validate data generation strategies before committing to expensive model training. The three evaluation dimensions (relevance, human-likeness, diversity) are comprehensive and well-reasoned.
4. Solid Downstream Task Validation: A key strength is that the paper does not stop at data-level evaluation (Table 1) but proceeds to validate the data's ultimate effectiveness through downstream NPC fine-tuning experiments (Table 2). The results strongly demonstrate that small models with high-quality data can approach the performance of large prompt-based NPCs, providing new evidence for the "data quality over model scale" argument, which is highly relevant to the industry.

**Weaknesses:**

1. Questionable Strength and Fairness of Baselines: The main baselines, "ICL" and "RPA," are "adapted" by the authors from SOTA methods. This raises a core question: are these baselines sufficiently strong?
  - For the RPA (Role-Playing Agent) baseline, the paper claims it "lacks behavioral constraints," leading to poor output. However, RPA methods themselves often rely on strong prompting. Did the authors attempt to strengthen the RPA baseline with more sophisticated prompt engineering (e.g., explicitly instructing the RPA's player agent to "be concise," "think and type like a real player")?
  - The current comparison gives the impression that the "Fake Player" architecture's advantage is overwhelming, but this might be contingent on weakly-implemented baselines.
2. Lack of Reliability Validation for "Distill Bench": "Distill Bench" is a core contribution, but it relies heavily on "LLM as a Judge" (using DeepSeek-V3) for automated evaluation. It is well-known that LLM-as-a-Judge methods suffer from biases (e.g., position, verbosity) and consistency issues.
  - The paper lacks a crucial validation experiment: What is the correlation between the "Distill Bench" automated scores and "human evaluation" scores?
  - If the Judge LLM were replaced (e.g., with GPT-4, Llama 3, or Claude 3), would the ranking of methods in Table 1 (Fake Player > RPA > ICL) remain consistent? Without this validation, the reliability of "Distill Bench" is questionable.
3. Missing Ablation Study for the "Typing Agent": The "Typing Agent" aims to simulate human typing behavior by introducing typos, grammatical errors, etc. This is an interesting detail, but what is its actual contribution to the downstream SFT task?
  - Intuitively, injecting "noise" (like typos) into SFT data might harm model learning rather than enhance it.
  - The paper lacks an ablation study for the "Typing Agent." If this module were removed and the "Intent Agent's" top intent $C_i$ were directly converted into concise, natural language (without errors), how much would the downstream NPC performance (Table 2) drop? Is this module "essential" or just "nice to have"?
4. Framework Complexity and Robustness: "Fake Player" uses a three-agent serial pipeline. This introduces complexity and a potential for "compounding errors"—a small deviation in the "Inner Monologue" could be amplified by the "Intent Analysis" and "Typing Agent." The paper does not discuss the framework's robustness, e.g., its sensitivity to the initial (P, D, T) seed settings.

**Questions:**

1. Can you elaborate on the implementation details of the RPA baseline? Specifically, did you attempt to strengthen the RPA's "player agent" prompt to simulate "conciseness" and "human constraints" to create a more competitive baseline?
2. Regarding the reliability of "Distill Bench." Have you conducted a correlation analysis between "Distill Bench's" automated scores and human expert ratings? Do the conclusions in Table 1 hold if you use a different LLM (e.g., GPT-4 or Claude 3) as the judge?
3. What is the specific contribution of the "Typing Agent" module? Have you run an ablation study where the "Typing Agent" is replaced with a simple "intent-to-text" converter (i.e., preserving conciseness but removing noise like typos)? How does this affect the downstream NPC performance in Table 2? Is intentionally introducing errors into SFT data truly beneficial?
4. How robust is the three-agent serial framework? For example, how sensitive is the iterative update in Equation (2) to the quality of the "Inner Monologue"? Is there a problem with error accumulation and amplification?

---

> ### Author Response · Authors · 2025-11-21
> **(1) Response to Reviewer CQsN**
>
> Thank you for reviewing our work. Below, we will address your concerns and suggestions point by point:
>
> # 1 Regarding the implementation details of the RPA baseline
>
> On the issue of baseline strength: The prompt templates used for the RPA baseline were specifically designed and adapted to fit our NPC companionship task. Since our task differs from traditional role-playing data scenarios, targeted adaptations were necessary to ensure alignment with our task objectives. The corresponding prompt code is provided in the Supplementary Material and has also been updated in the appendix of the latest manuscript.
>
> In fact, from the results in Table 2, the performance of NPC models fine-tuned on data distilled by RPA and ICL can be comparable to that of some large-parameter models—this directly demonstrates that these two baseline methods themselves have solid performance.
>
> The table below presents experiments on refining the RPA prompt. However, such refinement did not necessarily yield overall better results: although Human-Likeness was slightly improved, data diversity was significantly reduced. Human-Likeness and diversity are essentially mutually constrained dimensions, and our Fake Player method addresses this trade-off , enabling strong performance in both dimensions.
>
> *Methods** | **Topic Breadth** | **Topic Depth** | **Total (Topic Relevance)** | **Conciseness** | **Improvisation** | **Total (Human-Likeness)** | **Diversity**| **Overall** |
> | --- | --- | --- | --- | --- | --- | --- | :---: | :---: |
> | ICL | 72.5 | 73.3 | 72.9 | 59.40 | 74.60 | 67.00 | 46.63 | 62.18 |
> | RPA | 76.80  | 69.2 | 73.05 | 45.60 | 72.60 | 59.10 | **67.79** | 66.65 |
> | RPA_refine | 77.2 | 74.6 | **75.59** | 52.8 | 73.76 | 63.28 | 57.3 | 65.39 |
> | Fake Player | 73.2 | 75 | 74.1 | 64.40 | 75.20 | **69.8** | 67.28 | **70.39** |
>
> Additionally, overly refined prompts tend to overfit the specific target task. This means that when switching to other NPC-related tasks, significant time and effort would be required to re-polish the prompts. In contrast, Fake Player does not rely on excessive prompt refinement; instead, it achieves superior performance through simple prompts and an optimized pipeline design.
>
> # 2 Regarding the reliability Validation for "Distill Bench"
>
> Our original intention in designing Distill Bench is to establish an efficient automated evaluation process to support the rapid iteration of data distillation evaluation training. Therefore, we use LLM as judge as the core evaluation method, which can significantly reduce the dependence on manual annotation.
>
> In particular, the selection of this assessment method is based on rigorous preliminary verification. At the beginning of the study, we have systematically verified the effectiveness of the LLM as a judge evaluation criteria adopted in this paper through the manual evaluation of a large number of team members (including the author of the paper). After confirming that it is highly consistent with the results of manual evaluation, the iterative optimization of subsequent models establishes an efficient model based on "LLM as a judge, supplemented by a small amount of manual experience verification".
>
> In order to further prove the relevance of LLM evaluation and human evaluation, we designed a more extensive validation experiment: 50 multi-turn dialogue samples were randomly selected from each method (ICL, RPA, Fake Player), and five independent evaluators outside the team (all with graduate degrees, majors covering computer science and psychology, and in-depth understanding of the game field) were employed to score manually according to the criteria completely consistent with LLM evaluation. The experimental results are shown in the table below:
>
> | **Methods**   | **Topic Breadth** | **Topic Depth** | **Total (Topic Relevance)** | **Conciseness** | **Improvisation** | **Total (Human-Likeness)** |
> |---------------|-------------------|-----------------|------------------------------|-----------------|-------------------|-----------------------------|
> | ICL           | 66.2              | 72.2            | 69.15                        | 71.2            | 66                | 68.6                        |
> | RPA           | 72.4              | 67.6            | 70                           | 40.4            | 57.2              | 48.8                        |
> | Fake Player   | 75.2              | 70.4            | **72.8**                         | 74.4            | 72.4              | **73.4**                        |

---

> > ### Author Response · Authors · 2025-11-21
> > **(2) Response to Reviewer CQsN**
> >
> > The analysis shows that: the manual evaluation has a stronger ability to identify the sentences obviously generated by AI, which leads to the humanoid score of RPA method is significantly lower than that of LLM evaluation. However, in the key dimensions, the manual evaluation and LLM evaluation show a high degree of consistency: in terms of topic relevance, fake player performs best; In terms of pseudo humanity, the order of the three is FP>ICL>RPA. This result is completely consistent with the evaluation trend of using DeepSeek-V3 as the judge model.
> >
> > # 3 Regarding the Typing Agent
> > The core contribution of the Typing Agent lies in simulating real human players’ typing scenarios—it converts players’ internal thoughts (guided by the intent stack) into actual typed text. This design is derived from an insight gained through analyzing dialogue data in practical industrial scenarios: in Chinese-language game interactions, players typically use Pinyin input methods on mobile devices or keyboards, which inherently leads to a high probability of typos.
> >
> > To align with this real-user behavior, our Typing Agent specifically simulates such typo scenarios—for example, common input errors like confusing "大侠 (dà xiá, great hero)" with "大虾 (dà xiā, big shrimp)" (due to similar Pinyin spelling "dà xiā"), or mistyping "任务 (rèn wu, task)" as "认务 (rèn wu)" (a common Pinyin input error). By incorporating these realistic typos into training, we better align the synthesized data with actual player interaction habits.
> >
> > It is worth noting that this function serves as a "value-added optimization"—a targeted modeling for Chinese players. Since it primarily enhances the authenticity of text expression rather than altering core interaction logic, it does not have a significant impact on the downstream performance of NPC models.
> >
> > # 4 Regarding the robustness of our framework
> > Our method targets open-ended NPC companionship tasks, where the goal is to enable NPCs to provide high-quality responses to both casual chats and knowledge-seeking queries from players with diverse personalities. Thus, when distilling and generating dialogue data, our core focus is on producing player-NPC interaction data that balances authenticity and diversity.
> >
> > Notably, the "instability" in the interaction among our three agents (Inner Monologue Agent, Intent Analysis Agent, and Typing Agent) is precisely what drives the diversity of generated data. Unlike math problems that have a single correct answer, our task requires simulating a Fake Player to continuously initiate conversations and ask questions to NPCs—aiming to cover as many types of NPC responses as possible. Therefore, we strive to generate varied interaction trajectories from a single seed setting, which significantly reduces the cost of data synthesis.
> >
> > Take the ICL method as an example: If we use one seed setting to generate 10 dialogue trajectories repeatedly, the resulting data will suffer from insufficient diversity and a high risk of overfitting, leaving only 2–3 trajectories usable. In contrast, with our Fake Player framework, repeating the same seed setting 10 times yields 10 highly diverse data trajectories. This directly reduces the labor cost required for constructing additional seed settings, as a single seed can be leveraged to produce a large volume of valid, diverse interaction data—ultimately reflecting the strong robustness of our framework in adapting to open-ended companionship tasks and optimizing data efficiency.
> >
> > **Thanks for your review again! We hope our responses can address your concerns.**

---

### Official Review · Reviewer_mgWL · 2025-11-01

**Soundness:** 2
**Presentation:** 2
**Contribution:** 2
**Rating:** 2
**Confidence:** 4

**Summary:**

This paper introduces a new approach to modeling players in gaming settings that have them interact with NPCs. Players are modeled using a multistep process where players have their own longer inner monologue, which is turned into intents, and then these intents are turned into an utterance for an NPC. To test the setup, the NPCs are simulated with LLMs of various sizes, as well as smaller LMs that are trained on distilled interactions. Interactions are scored using LLM-as-judge with two criteria for human-likeness, conciseness and improvisation, and two for topical relevance, depth and breadth, as well as a third custom, data diversity score.

**Strengths:**

- Interesting and novel design for how to model player mental states while effectively producing realistic interactions with NPCs.

- Multiple evaluate metrics are considered to provide a more holistic assessment

- I was very excited to see the details in Appendix F that this system has been deployed. However, details on this deployment are very minimal---likely due to the non-disclosure. I wish the authors could have highlighted this more though, as it would immensely strengthen the paper.

**Weaknesses:**

- I found this paper surprisingly hard to follow. The paper does a good job of presenting high level information and narrative, but many specifics and details are hard to track down. One key missing piece is how these LLM based agents are actually designed and used. The prompts for the agents are not presented (only the evaluation prompts, as far as I can tell), so I am not sure how to evaluate the generality of the approach. Other examples include the presentation such as in the tables, the metrics are reported with two-letter abbreviations which, to me, were not easily identifiable without more searching.

- Other experimental details are hard to assess. I appreciate the authors comparing with other baselines like ICL and RPA. However the paper says that these are adapted from their original approach but little details are provided. For example, it's not clear how many shorts are used in ICL. It's also not clear how much of the core player guidelines are shared between all three of these approaches (which would be necessary for modeling players), which prevents assessing whether the improvement is due to better prompts or the agentic system.

- The paper makes extensive use of LLM as judge. I appreciate the authors using a different base model to evaluate to mitigate a model preferring its own outputs. However, these scores are never validated or compared against any kind of human output. The judge models are good, but some small part of the data needs to validated to calibrate how good they are at evaluating the qualities.

- The language in Lines 418-422 is talking about the relative distances of clusters in the t-SNE embedding, but t-SNE doesn't preserve global distances so I don't think these kinds of comparisons are meaningful. I think you'd want to use umap or PCA to substantiate these claims. I do appreciate the figure though.

**Questions:**

- The discussion of Bratman (1987) and player intent in lines 191-195 is very interesting for thinking about how to model players. Are there any citations to back up some of these claims?

- I am confused by the use of a stack in the intent agent. As far as I can tell, the stack operations in (3) always push one or more item based on $S_{i-1}$, and then pop the top intent. However, I don't see any way to actually move deeper into the stack (e.g., by popping more items off or viewing more). There's a vague statement about operations in 209 but I'm not sure if these operations would do so. If not, why not just keep the most recent state around and discard all earlier states?

- For topical relevance, how are topics defined?

- The data diversity scoring why are the operations in the inner for loop needed? For example, why couldn't you measure diversity by the average pair-wise embedding similarity for utterances---i.e., why must a probability distribution be made and entropy be calculated? I realize there are multiple ways to measure what you're doing and I'm not saying the approach is wrong, but I don't yet understand the intuition for why this complexity is needed.

- In line 319, the data is described as a "real-sampled resources". Could you say more about what is the "real" part?

- How were the win-rates produced in Table 4? Are these from people or LLMs?

- For the human likeness experiments in Appendix A, what are the settings for this human-NPC experiment? How many humans and how many unique NPCs? Are these players (real or simulated) all interacting with the same intents?

---

> ### Author Response · Authors · 2025-11-21
> **(1) Response to Reviewer mgWL**
>
> Thank you for reviewing our work. Below, we will address your concerns and suggestions point by point:
> # 1 Regarding the missing design and use of the LLM-based agents
>
> We sincerely apologize for the oversight in not explicitly stating that the detailed prompt templates were provided in the Supplementary Material. Our initial intention was to conserve manuscript space, as the prompts are quite lengthy. We have now updated the manuscript to clearly indicate that the complete prompt designs for all methods (including ICL, RPA, and our proposed Fake Player) are available in the Appendix for full transparency and reproducibility.
>
> Furthermore, to foster open research and community contribution, we will open-source the implementation of our Fake Player framework and the adapted baselines on GitHub. This method has been successfully deployed in industrial applications, serving over millions players, and we believe the released code and prompts will provide valuable reference for both academic research and practical deployments. We are confident that the detailed methodologies provided in the Appendix, coupled with the upcoming open-source code, will significantly aid in understanding and replicating our work.
>
> # 2 Regarding the comparing details
> To ensure a fair comparison, we intentionally used generic prompts for all methods, avoiding extensive optimization that could bias the results.
> Specifically, our ICL baseline employs few-shot learning with examples covering both key domains: Chit-Chat and Knowledge-based. Crucially, player modeling was standardized by using the same player_persona settings across ICL, RPA, and our Fake Player. This controlled approach ensures that performance differences stem from the methodologies themselves, providing a rigorous evaluation of their core capabilities in generating authentic interactions.
>
> # 3 Regarding the readability of table
> We have optimized the readability of Table 1 and Table 2 and updated the manuscript to facilitate readers' understanding. Thank you for bringing this up, which is very helpful for us
>
> # 4 Regarding the use of LLM as a Judge
> Our original intention in designing Distill Bench is to establish an efficient automated evaluation process to support the rapid iteration of data distillation evaluation training. Therefore, we use LLM as judge as the core evaluation method, which can significantly reduce the dependence on manual annotation.
> In particular, the selection of this assessment method is based on rigorous preliminary verification. At the beginning of the study, we have systematically verified the effectiveness of the LLM as a judge evaluation criteria adopted in this paper through the manual evaluation of a large number of team members (including the author of the paper). After confirming that it is highly consistent with the results of manual evaluation, the iterative optimization of subsequent models establishes an efficient model based on "LLM as a judge, supplemented by a small amount of manual experience verification".
> In order to further prove the relevance of LLM evaluation and human evaluation, we designed a more extensive validation experiment: 50 multi-turn dialogue samples were randomly selected from each method (ICL, RPA, Fake Player), and five independent evaluators outside the team (all with graduate degrees, majors covering computer science and psychology, and in-depth understanding of the game field) were employed to score manually according to the criteria completely consistent with LLM evaluation. The experimental results are shown in the table below:
>
>
> | **Methods**   | **Topic Breadth** | **Topic Depth** | **Total (Topic Relevance)** | **Conciseness** | **Improvisation** | **Total (Human-Likeness)** |
> |---------------|-------------------|-----------------|------------------------------|-----------------|-------------------|-----------------------------|
> | ICL           | 66.2              | 72.2            | 69.15                        | 71.2            | 66                | 68.6                        |
> | RPA           | 72.4              | 67.6            | 70                           | 40.4            | 57.2              | 48.8                        |
> | Fake Player   | 75.2              | 70.4            | **72.8**                         | 74.4            | 72.4              | **73.4**                        |
>
> The analysis shows that: the manual evaluation has a stronger ability to identify the sentences obviously generated by AI, which leads to the humanoid score of RPA method is significantly lower than that of LLM evaluation. However, in the key dimensions, the manual evaluation and LLM evaluation show a high degree of consistency: in terms of topic relevance, fake player performs best; In terms of pseudo humanity, the order of the three is Fake Player>ICL>RPA. This result is completely consistent with the evaluation trend of using DeepSeek-V3 as the judge model.

---

> > ### Author Response · Authors · 2025-11-21
> > **(2) Response to Reviewer mgWL**
> >
> > # 5 Regarding the t-SNE
> > Our core goal is to qualitatively show the distribution differences of chit chat type and knowledge-based type data generated by different methods, rather than quantitatively evaluate the global distance between clusters. By optimizing the preservation of local neighborhood relations, t-sne can efficiently highlight the "aggregation degree of similar data" and "separation trend of different types of data" in high-dimensional embedded space. This is the core of our verification: whether the data generated by fake player presents more scattered clustering (reflecting diversity) under the same seed setting, and whether the clustering of different domains (chitchat/knowledge) can form a clear distinction (reflecting domain adaptability).
> >
> > In the experimental results, the tight clustering of ICL, the clustering defects of RPA in the field of chitchat, and the decentralized and well-defined clustering mode of fake player in both fields are fully consistent with our visualization goals, effectively supporting the conclusion that "the diversity of data generated by fake player is better".
> >
> > Secondly, we fully recognize the feature that t-sne does not retain the global distance, but in the visualization scenario of this study, this feature does not affect the validity of the conclusion. On the one hand, our quantitative diversity assessment has been implemented by the "text embedding similarity+information entropy" method of algorithm 2, and t-sne is only used as a supplementary qualitative evidence, not a core quantitative basis.
> >
> > # 6 Regarding the citations to back up some of Intent Stack
> >
> > Thank you for your interest in this aspect! Regarding the modeling of human intent, the field of Intent Classification in Spoken Language Understanding (SLU) has accumulated rich relevant research. For instance, Huang et al. [1] investigated multi-turn dialogues between telecom users and customer service representatives, focusing on how to truly grasp the core intent behind users’ calls through hierarchical intent modeling.
> >
> > In real-world communication scenarios like customer service consultations, users typically initiate interactions with a clear primary intent. However, during the dialogue process, their fine-grained intents often evolve dynamically in response to context—resulting in utterances that convey distinct intent nuances at different turns. This aligns perfectly with the behavioral characteristics of human communication we emphasized, as well as the design rationale of the intent stack in our work, which aims to capture such dynamic intent shifts.
> >
> > [1] S. Huang, et al. Exploring Label Hierarchy in Dialogue Intent Classification. ICASSP2024
> >
> > # 7 Regarding the use of the intent stack in the intent agent.
> > We will embed this stack into the prompt for the agent to reference. The agent will then update the stack based on its understanding, with the update operations including deletion, addition, merging, and no-operation, as specified in line 209.
> >
> > # 8 Regarding the definition of topics for topical relevance evaluation
> > In the field of game intelligent NPCs, distinguishing between Chit Chat (casual chat) and Knowledge-based (knowledge-seeking) dialogues is a widely recognized categorization for companion AI NPCs. Building on this industry consensus, we further define topics as encompassing these two core categories and their derived sub-topics. Specifically, we sample and synthesize additional sub-topics from real human dialogue data to enrich the topic system.
> >
> > A concrete example of our topic definition is provided in the supplementary material, as shown below:{"Topic": "Relationship", "Sub-topic": "Healing from a Broken Heart", "Chat Direction": "When looking at old photos late at night, you smell the lingering perfume on a shirt and the hum of the refrigerator sounds unusually harsh. How did you get through the initial phase of a breakup?", "First Question Recommendation": "How long does it take to completely forget someone?"}

---

> > > ### Author Response · Authors · 2025-11-21
> > > **(3) Response to Reviewer mgWL**
> > >
> > > # 9 Regarding the data diversity scoring
> > > Thank you for raising this important point about the complexity of our diversity metric. We agree that simplicity is valuable, and our design choice was driven by the specific challenges of evaluating multi-turn dialogue data. Please allow us to explain the intuition behind our approach.
> > >
> > > First, multi-turn dialogues cannot be treated as a simple concatenation of independent turns. Each turn in a conversation has a distinct semantic structure and contributes differently to the overall diversity. A straightforward average of pairwise similarity across all turns would flatten these nuanced dynamics. Our method, which processes each turn individually in the inner loop, is designed to respect this hierarchical structure. It ensures that the diversity score accurately reflects the variation present at each stage of the conversation, which is crucial for training robust NPCs.
> > >
> > > Secondly, we found that using raw similarity scores as a direct proxy for diversity is problematic. It operates on a flawed assumption that "less similar" always equals "more diverse." In practice, an very low average similarity could indicate that the dialogues are random and off-topic, which is undesirable. True, valuable diversity exists within a consistent thematic context. This is why we introduced the information entropy calculation. By constructing a binary distribution from the maximum similarity per turn, we shift the focus from the absolute similarity value to the distribution of variations within the batch. Entropy naturally quantifies the uncertainty in this distribution, effectively penalizing batches that contain redundant pairs (low entropy) while rewarding those with healthy, meaningful variation (high entropy), even if the overall average similarity is moderate.
> > >
> > > In summary, the additional step of calculating entropy is not unnecessary complexity but a targeted solution. It allows our benchmark to prioritize the avoidance of repetition and the promotion of coherent diversity, which directly correlates with the quality of data needed for fine-tuning NPCs. We hope this clarification explains the rationale for our metric design.
> > >
> > > # 10 Regarding the real-sampled resources in the benchmark
> > > The personas and topics included in our benchmark are derived from real-sampled data—specifically, we conducted research and analysis on players across various games to summarize authentic user profiles and high-frequency topics that reflect real player behaviors and interaction demands.
> > > To further contribute to the community’s development, we are also willing to open up a portion of these personas and topics in subsequent work, facilitating more research on LLM-based NPC training and evaluation.
> > >
> > > # 11 Regarding the win rate experiment
> > > The win rate was calculated based on 100 sets of questions. Specifically, we used DeepSeek-V3-250324 as the judge model, which evaluated and compared the responses of different NPCs to the same questions. The final win rate was derived from the judge model’s assessments of which NPC’s response was superior for each question.
> > >
> > > # 12 Regarding the experiment in Appendix A
> > > The purpose of this experiment is to verify the reliability of the LLM judge method for the Human-Likeness dimension—specifically, to confirm whether responses more similar to human speech would receive higher scores. To this end, we invited 35 human players to conduct multi-turn dialogues with the same NPC, then evaluated the human players’ utterances using the same LLM evaluation criteria applied to fake players. The results showed that human players’ utterances indeed achieved the highest scores. It is worth noting that both real players and fake players were provided with the same designated dialogue topics to ensure the consistency of the evaluation context.
> > >
> > > # 13 Regarding the industry deployment
> > > We have supplemented detailed information about our industry deployment in the appendix F, which specifically includes the NPC architecture and deployment strategies that we actually adopted in practical application scenarios. To our knowledge, the architecture and strategies we used are also relatively standard processes in the industry, having been applied to many game intelligent NPC projects and emotional companion products.
> > > Notably, there is relatively little public information available on such practical industry deployment details for LLM-based NPCs. We believe the supplementary content can provide valuable references for other teams with similar deployment needs. If the reviewers have concerns about additional deployment details, we are happy to provide further explanations or materials.
> > >
> > > **Thanks for your review again! We hope our responses can address your concerns.**

---

### Author Response · Authors · 2025-11-28
**General response to reviewers**

Dear Reviewers and Area Chairs,

As the rebuttal deadline approaches, we sincerely invite you to take a moment to review our responses. Below, we also provide some general clarifications on points raised by multiple reviewers:

# 1. Regarding the implementation details of baselines and the fake player
These were initially provided in the Supplementary Material and have now been updated in the appendix of the revised manuscript. We also plan to release the code publicly to contribute to the community.

# 2. Regrading the human evaluation
Our use of "LLM as a judge" is strongly justified by its high consistency with human evaluation. To further validate this, we conducted a rigorous experiment: Five independent evaluators (all holding graduate degrees in computer science or psychology, with a deep understanding of the gaming domain) manually scored 50 randomly selected multi-turn dialogue samples from each method (ICL, RPA, Fake Player), using the exact same criteria as the LLM evaluation. The results are summarized below:

| **Methods**   | **Topic Breadth** | **Topic Depth** | **Total (Topic Relevance)** | **Conciseness** | **Improvisation** | **Total (Human-Likeness)** |
|---------------|-------------------|-----------------|------------------------------|-----------------|-------------------|-----------------------------|
| ICL           | 66.2              | 72.2            | 69.15                        | 71.2            | 66                | 68.6                        |
| RPA           | 72.4              | 67.6            | 70                           | 40.4            | 57.2              | 48.8                        |
| Fake Player   | 75.2              | 70.4            | **72.8**                         | 74.4            | 72.4              | **73.4**                        |

The strong alignment between human and LLM evaluations demonstrates the reliability of our approach.


# 3. Regrading the Data Generation Costs
We quantified the cost and latency of Fake Player vs. baselines using DeepSeek-V3-250324 (cost: 2 CNY/1M input tokens, 8 CNY/1M output tokens; 1 USD ≈ 7.14 CNY). For 500 multi-turn samples (concurrency=50):

| Method | Input Token | Input Cost (CNY) | Output Token | Output Cost (CNY) | Total Cost (CNY) | Total Cost (US Dollar) | Time Consumption (s) |
| --- | --- | --- | --- | --- | --- | --- | --- |
| ICL | 4106549 | 8.21 | 246121 | 1.97 | 10.18 | 1.43 | 100 |
| RPA | 5834138 | 11.67 | 324212 | 2.59| 14.26| 1.99 | 402 |
| Fake Player | 6231786 | 12.46 | 574966 | 4.60 | 17.06 | 2.39 | 596 |

For reference, manually annotated multi-turn dialogue data costs an average of 5 CNY per sample—generating 500 samples would cost 2500 CNY (≈ 350 US Dollars), which is far higher than the costs of all three automated methods.

While Fake Player’s synthesis cost is slightly higher than that of ICL and RPA, it still remains in a highly cost-effective range (and is vastly cheaper than manual annotation). More importantly, ICL and RPA fail to meet the quality requirements for product deployment due to issues like insufficient diversity or poor human-likeness; in contrast, **Fake Player achieves production-ready data quality** with its balanced performance in authenticity, diversity, and topic relevance. This high-quality data further helps save downstream NPC model training costs (e.g., reducing iterations caused by low-quality data).

Notably, for in-development games planning to deploy large-scale NPCs, Fake Player’s cost-effectiveness will be even more prominent: its ability to generate a large volume of high-quality data from minimal seed settings avoids the exponential labor and time costs of manual annotation or repeated prompt polishing for other methods.


**We thank the reviewers for their thoughtful and constructive feedback. We look forward to your response.**

---

### Author Response · Authors · 2025-11-29
**Summary of Contributions and Clarifications**

Dear Area Chairs,

Thank you for handling our submission. **In our previous rebuttal, we have also clarified several misunderstandings raised by the reviewers and provided the additional experiments and explanations they requested.** We respectfully summarize the key strengths of our paper “Fake Player: Imitating Real Players to Distill Data for LLM-based NPC Training”, as consistently acknowledged across reviewers, to highlight its contribution to the ICLR community.

# 1. Addresses a Critical and Previously Overlooked Research Gap

Existing research overwhelmingly focuses on NPC-side persona alignment, leaving the player-side behavior space severely under-modeled. This gap is crucial: real player inputs directly determine NPC robustness, emotional intelligence, and immersion. Without realistic player-side data, even well-designed NPCs fail in deployment.

Our work is the first to explicitly address this missing half of the alignment pipeline.
Filling this gap transforms data generation for LLM agents—not only for games but broadly for any interactive chat-based agent requiring human-side simulation.

# 2. Novel and Insightful Multi-Agent Framework Modeling Human Cognitive-to-Expression Pipeline

Reviewers highlighted the Fake Player architecture as a conceptually original contribution.
The three-agent design—Inner Monologue, Intent Analysis with an intention stack, and Typing Agent simulating real-world expression constraints—offers a mechanistic and cognitively grounded way to generate human-like player behavior, surpassing naive role-play or few-shot approaches.

This framework originated from large-scale analysis of actual user interaction logs.
Because it abstracts human behavior into generalizable cognitive stages, it is applicable not only to games, but to any LLM scenario needing realistic, concise, human-like user-side inputs (e.g., customer service, social chatbots, tutoring dialogue systems).

**In addition, we supplemented the paper with new experiments and theoretical explanations, which specifically resolved the concerns raised by reviewer mgWL and reviewer CQsN.**

# 3. Practical and Valuable Benchmark for Low-Cost Data Quality Evaluation

The proposed Distill Bench was recognized as an important engineering contribution, enabling quantitative, pre-training evaluation of distilled data along relevance, human-likeness, and diversity dimensions. This design substantially reduces iteration cycles in industrial LLM-NPC development.

Moreover, both our extensive internal expert verification and subsequent external human evaluation demonstrate the reliability of this benchmark. These validations directly address reviewers’ concerns regarding the consistency between human assessment and LLM-based judgments.

Finally, the benchmark also lays the groundwork for future research toward more automated, scalable data-quality evaluation, making it a practically valuable tool for the community.

# 4. Strong Downstream Evidence: Data Quality Beats Model Scale

Experiments show that small 7B models fine-tuned on Fake Player data can match or exceed much larger prompt-based models.
Reviewers recognized this as strong empirical validation and as evidence supporting an emerging paradigm: high-quality data can compensate for smaller models—a conclusion important to both research and applied settings.

To the best of our knowledge, this is the first industrial deployment where an NPC system reaches production quality solely from synthetic data without any human-annotated dialogue.
This demonstrates an important paradigm shift: data quality—not model size—is the key bottleneck in real-world agent deployment.


# 5. Real-World Deployment Demonstrates Impact and Feasibility

Reviewers explicitly appreciated that our system has been deployed industrially, confirming both practical relevance and the robustness of the proposed design.

Additionally, we provide deployment details (Appendix F), offering rare, practical insights for both academic and industrial communities.
This real-world validation shows our method is not only theoretically compelling but also highly feasible in practice.

----
We hope this summary clarifies why the reviewers acknowledged our submission as a novel, high-value contribution with solid empirical and practical impact. We believe the paper offers meaningful advances to the research community.

Thank you again for your consideration.

Sincerely,

The Authors

---

### Meta-Review · Area_Chair_9EYQ · 2026-01-06

**Summary:**

This paper proposes the "Fake Player" framework, which employs multi-agent collaboration to simulate real player behavior, thereby distilling high-quality, diverse dialogue data for fine-tuning NPC models in games. Additionally, the paper designs "Distill Bench" to evaluate the quality of the synthesized data.

**Reviewer Concerns:**

- **Reviewer mgWL: [Score 2]** Strongly criticized the paper’s readability, lack of baseline details, and the erroneous t‑SNE analysis, concluding that the paper in its current state is not suitable for publication.
- **Reviewer CQsN: [Score 4]** Expressed concerns about baseline strength and evaluation reliability. While the rebuttal provided additional data, worries about the system’s complexity were not fully resolved.
- **Reviewers PUtu & NmYs: [Score 6]** Were relatively positive, focusing more on cost and application value, but their examination of methodological details was not as in‑depth as mgWL’s.
- **Main Point of Divergence**: The other reviewers concentrated more on application‑level effectiveness, while reviewer mgWL identified fundamental shortcomings in scientific rigor and presentation.

**Reviewer Scores:**

Although the "player simulation" concept proposed in this paper possesses some novelty, the paper exhibits significant weaknesses in execution quality. The rebuttal supplemented some data and explanations but could not remedy the disorganized writing structure and methodological imprecision in the main text within the limited time. Furthermore, the fairness of baseline comparisons still lacks sufficient transparency to dispel doubts. To uphold the rigor expected of conference papers, rejection is recommended.

---

### Decision · Program_Chairs · 2026-01-26

Reject